# PI-Controlled Uncertainty for Steady-State Error Elimination in Ultrasound Image Segmentation

## Abstract

Accurate segmentation of anatomical structures from medical ultrasound images is essential for reliable diagnosis, yet conventional training losses often leave persistent steady-state errors, especially along ambiguous boundaries. These losses act as control variables generated by a proportional controller, since they respond only to instantaneous discrepancies and lack the memory required to correct long-term deviations. To overcome this limitation, we rethink segmentation training as a closed-loop control system where uncertainty acts as the control variable. Building on this perspective, we introduce a proportional–integral (PI) control mechanism that integrates both present and historical error signals into the optimization process, enabling the model to systematically eliminate steady-state errors and deliver sharper, more reliable boundary predictions. Unlike existing uncertainty-based approaches that rely solely on fixed loss terms, our method provides a principled mechanism to incorporate dynamic feedback into training. The framework is model-agnostic and introduces no additional inference overhead, making it directly compatible with real-time segmentation backbones. Extensive experiments on clinical medical ultrasound datasets demonstrate consistent improvements over state-of-the-art baselines. These results confirm that our framework offers an effective solution for eliminating steady-state errors in medical ultrasound image segmentation under challenging conditions. Our code is available at `https://anonymous.4open.science/r/PI-control-uncertainty-B82C`.

## 1 Introduction

Accurate semantic segmentation of anatomical structures, particularly from ultrasound imaging, provides crucial quantitative support for clinical decision-making, including disease diagnosis, treatment planning, and prognostic monitoring (Azad et al., 2024; Tiwari et al., 2025; Zhang et al., 2020). A large number of learnable methods have achieved significant results in recent years (Wang et al., 2021; Li et al., 2025; Hu et al., 2025). Nevertheless, due to the scattering and attenuation characteristics of ultrasound waves in tissues, medical ultrasound images suffer from more severe low contrast, speckle noise, and ambiguous tissue boundaries compared to CT and MRI (Lee et al., 2022; Zamzmi et al., 2021; Gowda & Clifton, 2025). These image quality issues lead to higher uncertainty in segmentation tasks, making it crucial to learn uncertainty and leverage uncertainty to improve segmentation performance.

Methods for quantifying predictive uncertainty have been explored to enhance model reliability (Zhou et al., 2024; Mucsányi et al., 2024; Abdar et al., 2023; Judge et al., 2023; Zhou et al., 2025). They typically rely on static loss functions to learn uncertainty, for instance, by predicting the parameters of a probability distribution (Liu et al., 2022; He et al., 2019; Dong et al., 2025; Duenias et al., 2025). Unfortunately, this paradigm leads to a critical failure mode: the emergence of a steady-state segmentation mask prediction error, where a persistent discrepancy between the model's prediction and the ground truth remains, even after training converges. We identify that this limitation arises because conventional loss functions are analogous to simple proportional (P) controllers in classical control theory (Franklin et al., 2010; Åström & Hägglund, 2006). They address only the instantaneous error of each training step without memory of past failures, rendering them incapable of eliminating systematic, steady-state errors.

To address the above problems, We rethink existing uncertainty learning mechanisms through a closed-loop control framework. We analyze and reveal key insights into why P-control-like loss functions fail and how a more sophisticated control strategy can succeed. Inspired by these insights, we design a novel framework where the predicted segmentation mask is the controlled variable, driven to match the ground truth setpoint. Crucially, we define the uncertainty itself as the control variable, which is actively manipulated by our proposed controller to achieve precise error correction. Although some works, like PIDNet (Xu et al., 2023), incorporate PID control into network architecture to reduce feature fusion overshoot, their loss functions remains static. By training with static loss functions that capture only instantaneous error, these approaches inherently lack memory of historical information and cannot address the fundamental issue of steady-state error.

Furthermore, we introduce a Proportional-Integral (PI) controller as the core of our framework. Unlike P-control-like static losses, our PI controller leverages a crucial integral term to accumulate historical errors for persistently challenging regions. In this way, our controller can generate an adaptive and escalating correction specifically targeted at stubborn, long-term deviations. Therefore, our framework can effectively eliminate the steady-state error that plagues conventional methods, ensuring the model converges to a more accurate solution. In addition, our method can be seamlessly integrated into various segmentation backbones without extra inference cost.

Our main contributions are as follows: ❶ We rethink segmentation optimization as a control problem, applying uncertainty control directly to learning dynamics rather than conventional architectural design. This framework uniquely defines the segmentation mask as the controlled variable and the uncertainty as the control variable, offering a new lens for resolving persistent training errors. ❷ We design and implement a PI controller that integrates historical error information into the optimization process. This controller generates a dynamic, escalating corrective signal to precisely and effectively eliminate the steady-state errors that traditional methods fail to address. ❸ We evaluate our method on two distinct ultrasound datasets with different challenges: MEIS with blurred boundaries and TN3K with variable nodule characteristics, demonstrating significant performance improvements over state-of-the-art approaches.

## 2 RELATED WORKS

Foundational models like YOLACT (Bolya et al., 2019) and its successor for ultrasound, RAMEM (Tseng et al., 2024), provide efficient architectural baselines. Performance is further pushed by enhancing network components, such as introducing explicit boundary operators (Lin et al., 2023), transformer-based designs (Pei et al., 2022), advanced decoders (Wazir & Kim, 2025), or specialized loss functions to handle data imbalance (Xu et al., 2025). Concurrently, uncertainty quantification is often addressed through Bayesian methods (Gal & Ghahramani, 2016) or Deep Ensembles (Lakshminarayanan et al., 2017). Control theory is also applied, primarily to network architecture design (Girum et al., 2021; Xu et al., 2023) or as an inspiration for optimizers An et al. (2018). Further details on related works are provided in Appendix A.4.

However, these approaches share a critical limitation: they are all optimized using static losses that lack a dynamic mechanism to correct for the steady-state errors that persist during training. Integral control is a mechanism specifically designed to eliminate persistent errors by accumulating historical information. However, while some methods apply control theory to network structure, its potential remains underexplored in the context of the optimization process itself. Our work identifies this fundamental gap and proposes a PI-controlled uncertainty mechanism to directly address stubborn boundary inaccuracies by rethinking the training dynamics.

## 3 METHOD

### 3.1 PROBLEM FORMULATION

Different image segmentation methods use various neural network outputs to represent uncertainty, learning from the discrepancy between model predictions ($M_{\text{pred}}$) and ground truth ($M_{\text{gt}}$). At each step, the loss function calculates the error $e(t) = |M_{\text{gt}} - M_{\text{pred}}|$, and the optimizer updates model parameters using the gradient to minimize this error. From a control theory perspective, this resembles a feedback control system where the corrective signal is proportional to the instantaneous error

$e(t)$, similar to a Proportional (P) controller $u(t) = K_p \cdot e(t)$, where $K_p$ is the proportional gain. The comparison between conventional learning processes and our PI-control framework is shown in Fig. 1.

The fundamental deficiency of this P-control-like mechanism is its inherent inability to eliminate systematic steady-state error. To illustrate this limitation, consider medical image segmentation with ambiguous tissue boundaries. After training, a model's prediction for a ambiguous pixel may converge to 0.55 probability despite a ground truth of 1, yielding a constant error $e(t) = 0.45$. This persistent error has negligible influence on the loss function minimization compared to the much larger number of easily-classified pixels. Consequently, the optimizer provides insufficient updates to resolve this deviation, resulting in steady-state segmentation mask prediction error. Standard training paradigms, behaving like P-controllers, cannot eliminate such persistent errors due to their reliance solely on instantaneous error signals without error accumulation mechanisms.

### 3.2 OVERALL FRAMEWORK

To resolve the steady-state error in segmentation tasks, which arises from conventional loss functions, we reframe the training process of deep neural networks as a closed-loop feedback control system, as shown in Fig. 1. The core of this framework is a PI controller that acts as a dynamic construction mechanism, generating the final control variable by integrating a modulated error signal with the model's predictive uncertainty. It is composed of the following key components.

#### 3.2.1 CONTROLLED OBJECT AND FEEDBACK

Within our control system, the controlled object is the model itself. Our work is built upon an existing framework, RAMEM, which is based on the real-time instance segmentation model, YOLACT, demonstrates impressive performance in M-mode echocardiography. A detailed description of the RAMEM is provided in Appendix A.2. It is composed of three main components: a backbone, a Feature Pyramid Network (FPN), and the prediction heads. The backbone utilizes UPANet for feature extraction. The prediction heads consist of two primary sub-networks: the Prediction Module and the ProtoNet.

The Prediction Module predict a set of linear combination coefficients for each potential object instance. Concurrently, the ProtoNet generates a series of prototype masks. The final segmentation mask for each instance, denoted as $M_{pred}$, is then produced by linearly combining these prototype masks with the corresponding instance's coefficients. $M_{pred}$ is the system's Controlled Variable and is compared with the Ground Truth Mask $M_{gt}$, which acts as the Setpoint. This comparison yields an error signal, $e(t) = |M_{gt} - M_{pred}|$, quantifying the model's performance deviation and is fed back into our PI controller.

#### 3.2.2 PI-CONTROLLED UNCERTAINTY

The raw error signal $e(t)$ is fed into our designed PI controller. Unlike a classic PI controller that only processes an error signal, our proposed controller is an integrated module designed to generate the final control variable $L_{uncertainty}$ by simultaneously considering historical error dynamics and the model's predictive uncertainty. It consists of four components: the proportional term, ensuring a rapid response to current changes; the integral term, dedicated to eliminating long-term systemic biases; an Uncertainty Estimation Module, which predicts the uncertainty scale parameter $b$ for each instance; and a Laplace distribution to output the control variable $L_{uncertainty}$.

The proportional term focuses on the current instantaneous error. To make the controller more efficient, instead of using a global average error, we focus its attention on the model's most uncertain hard pixels, which is analogous to a form of online hard example mining. We first compute the per-pixel predictive uncertainty (measured as the variance of a Bernoulli distribution), $U = 4 \cdot M_{pred} \cdot (1 - M_{pred})$, which is maximized when the predicted probability $M_{pred}$ is close to 0.5. We then select the top-$k$ pixels with the highest uncertainty to form a hard set $\mathcal{H}$. This strategy prevents the controller's decision from being diluted by a large number of easily segmented background pixels, thereby concentrating its efforts on solving the real challenges. The proportional term $P(t)$

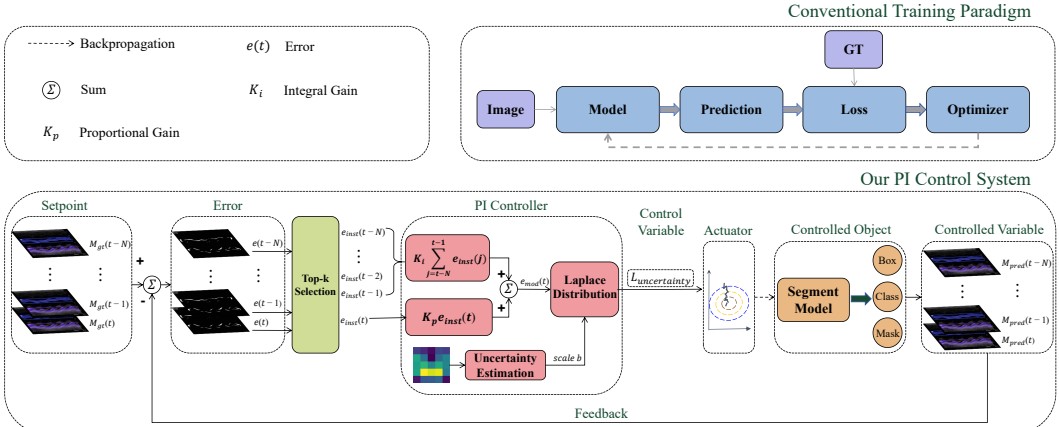

Figure 1: **Overview of the Proposed PI Control Framework.** This figure contrasts the conventional training paradigm with our control system approach. **Top**: The conventional training process, where the loss is driven only by the instantaneous error. **Bottom**: Our proposed PI Control System. This closed-loop framework introduces a novel PI controller as its core. An error signal, calculated as the difference between the ground truth (Setpoint) and the model's predictions over time, is processed by the controller. The controller internally performs (1) top-k selection on hard pixels to compute the proportional error term, (2) summation of historical errors for the integral term, and (3) learns an uncertainty parameter b via a dedicated estimation module. These components are then synthesized using a Laplace distribution to formulate the final $L_{\text{uncertainty}}$ (control variable). This signal is used by the optimizer (Actuator) to update the segmentation model (Controlled Object), effectively minimizing persistent steady-state errors.

is defined as the mean error over these hard pixels

$$P(t) = e_{\text{inst}}(t) = \frac{1}{k} \sum_{p \in \mathcal{H}} \left| M_{\text{gt}}^{(p)} - M_{\text{pred}}^{(p)} \right|, \tag{1}$$

where $e_{\text{inst}}(t)$ represents the instantaneous instance error for the current $t$, calculated as the average discrepancy over the set of $k$ most uncertain pixel indices $p$ within the hard set $\mathcal{H}$.

The integral term is the key to eliminating steady-state error, as it introduces memory into the system. In classical continuous control theory, the integral action is defined by the integral of the error signal over time: $I(t) = \int_0^t e(\tau)d\tau$. This mechanism ensures that any persistent, non-zero error, no matter how small, will eventually produce a significant corrective action. Since the training of a deep neural network is a discrete-time process that proceeds in epochs, we approximate this continuous integral with its discrete counterpart: a summation of errors over a finite time window. This provides the system with a practical "memory" of recent performance. Specifically, for each instance in the training set, we maintain a deque $D$ of length $N$ to store its average instance error, $e_{\text{inst}}(t), e_{\text{inst}}(t-1), ..., e_{\text{inst}}(t-N)$, over the past $N$ training epochs. The hyperparameter $N$ determines the system's memory length; a smaller value makes the controller more responsive to recent trends, while a larger value provides a more stable estimate of long-term systemic bias. The integral term $I(t)$ is defined as the sum of these historical errors

$$I(t) = \sum_{j=t-N}^{t-1} e_{\text{inst}}(j), \tag{2}$$

where $t$ represents the $t$-th epoch.

The final modulated error signal $e_{\text{mod}}(t)$ is the weighted sum of the two terms $e_{\text{mod}}(t) = K_{\text{p}} \cdot P(t) + K_{\text{i}} \cdot I(t)$, where $K_{\text{p}}$ and $K_{\text{i}}$ are the proportional and integral gain hyperparameters, respectively. When an instance's error persists, $P(t)$ may not change significantly, but $I(t)$ will accumulate, causing $e_{\text{mod}}(t)$ to escalate and form a targeted, ever-increasing corrective signal.

The controller's output, $e_{\text{mod}}(t)$, along with an uncertainty scale parameter $b$ predicted by the Uncertainty Estimation Module, is used to dynamically construct a supplementary loss term, which we call the PI-Controlled Uncertainty Loss ($L_{\text{uncertainty}}$). The Uncertainty Estimation Module consists of a 3x3 convolutional layer followed by a softplus activation function. The convolutional layer learns to extract features relevant to uncertainty from the shared feature map, while the softplus activation ensures that its output, the scale parameter $b$, is strictly positive, as mathematically required for Laplace distributions.

In our framework, this $L_{\text{uncertainty}}$ is defined as the system's control variable, $u(t)$. It is dynamically constructed by modeling the PI-modulated error signal $e_{\text{mod}}(t)$ with a Laplace distribution. The choice of the Laplace distribution over the more common Gaussian distribution is motivated by its heavier tails and sharper peak, which make it more robust to handling the potential outliers that can be generated by the PI controller's integral term, especially when $e_{\text{mod}}(t)$ becomes large. The general probability density function (PDF) of a Laplace distribution is given by

$$f(x \mid \mu, b) = \frac{1}{2b} \exp\left(-\frac{|x - \mu|}{b}\right), \tag{3}$$

where $\mu$ is the location parameter (mean) and $b$ is the scale parameter, which $2b^2$ the variance of the distribution. We set the mean $\mu$ to zero, as our goal is to drive the error to zero, and we treat our modulated error $e_{\text{mod}}(t)$ as a sample $x$ drawn from this distribution.

For a batch containing $M$ positive instances, we assume the modulated errors for each instance are independent. The model is trained to predict a unique scale parameter $b^{(i)}$ for each instance $i$, which quantifies the uncertainty. The joint PDF for observing the set of modulated errors $\{e_{\text{mod}}^{(i)}\}_{i=1}^{M}$ is the product of the individual Laplace PDFs. With $\mu = 0$, this becomes

$$f(\{e_{\text{mod}}^{(i)}\}_{i=1}^{M}, \{\mathbf{b}^{(i)}\}_{i=1}^{M}) = \prod_{i=1}^{M} \frac{1}{2b^{(i)}} \exp\left(-\frac{|e_{\text{mod}}^{(i)}(t)|}{b^{(i)}}\right). \tag{4}$$

To train this probabilistic model, we maximize the likelihood of the observed data, which is equivalent to minimizing the Negative Log-Likelihood (NLL). Taking the negative logarithm of the joint PDF gives us our final loss formulation for $L_{\text{uncertainty}}$

$$L_{\text{uncertainty}} = -\log(f) = \sum_{i=1}^{M} \left(\log(2b^{(i)}) + \frac{|e_{\text{mod}}^{(i)}(t)|}{b^{(i)}}\right). \tag{5}$$

This uncertainty-based modeling mechanism is ingeniously reflected in the loss function's design. The scale parameter $b^{(i)}$ serves as a direct indicator of the model's confidence. This relationship is mathematically grounded, as the entropy of the Laplace distribution is $1 + \log(2b)$. A larger scale parameter corresponds to higher entropy and thus lower model confidence. This creates a dynamic trade-off during training. On one hand, a larger $b^{(i)}$ value reduces the penalty from the position deviation term $|e_{\text{mod}}^{(i)}(t)|/b^{(i)}$, providing more lenient error tolerance for predictions with high, persistent uncertainty. On the other hand, the $\log(2b^{(i)})$ term acts as a regularizer that penalizes excessive uncertainty, preventing the model from simply increasing $b$ to ignore all errors. The pseudo-code of our PI-Controlled Uncertainty Learning mechanism is presented in Algorithm 1 in Appendix A.3.

The $L_{\text{uncertainty}}$ is backpropagated through an optimizer, which can be viewed as the Actuator. The optimizer calculates gradients and updates the weights of the model. This update alters the model's output $M_{\text{pred}}$, which in turn generates a new error signal in the next iteration, thus forming a closed control loop.

### 3.3 OVERALL LOSS FUNCTION

The final training loss function is a weighted sum of the original model RAMEM's losses and our newly introduced PI-controlled uncertainty loss. We intentionally retain the static mask loss ($L_{\text{mask}}$) alongside our new loss term, as they perform distinct and complementary roles. The standard static mask loss, a combination of BCE and Dice loss, is crucial for efficiently learning the features of the segmentation target, especially in the early stages of training.

In contrast, our PI-controlled uncertainty loss, $L_{\text{uncertainty}}$ , acts as a segmentation rectifier. Its primary role is to deal with the small subset of hard pixels that cause the steady-state error, which the static mask loss is ill-equipped to handle. To ensure training stability, we adopt a two-stage strategy where the $L_{\text{uncertainty}}$ term is introduced only after the preliminary training with mask loss reaches an initial feasible state. This allows the model to first learn the basic features of the task under the guidance of the mask loss before our PI controller begins its fine-grained correction process.

The overall loss function for end-to-end training is thus formulated as

$$L_{\text{total}} = \lambda_{\text{mask}} \cdot L_{\text{mask}} + \lambda_{\text{uncertainty}} \cdot \frac{1}{M} L_{\text{uncertainty}} + \lambda_{\text{cls}} \cdot L_{\text{cls}} + \lambda_{\text{bbox}} \cdot L_{\text{bbox}}, \quad (6)$$

where $M$ is the number of positive instances in the batch, and $\lambda$ terms are the respective loss weights. It is worth noting that our task is formulated as instance-level lesion segmentation rather than purely semantic segmentation. Since our framework is built upon the RAMEM instance segmentation pipeline, we retain the original classification and bounding box regression losses ($L_{\text{cls}}$ and $L_{\text{bbox}}$) from the baseline. Our PI-controlled uncertainty loss is only attached to the mask branch and does not alter the detection components, ensuring that the underlying instance segmentation architecture and inference procedure remain unchanged. By integrating our PI-controlled loss as an additional component, our method acts as an rectifier to the standard training process. This dynamic adjustment also provides valuable guidance for feature learning, directing the model to focus more on reliable features while being cautious with uncertain ones. This enhancement is achieved without adding any computational cost at inference time, as the PI controller and uncertainty loss are only active during training.

## 4 EXPERIMENT

We conduct experiments on two ultrasound datasets to validate our proposed PI-controlled training framework against multiple state-of-the-art methods. Our experimental evaluation is designed to answer the following key questions. Q1: Does the proposed PI-controlled framework outperform existing state-of-the-art methods in medical ultrasound segmentation? Q2: Can the proposed PI controller reduce segmentation errors on ambiguous boundaries compared to conventional methods? Q3: What are the individual contributions of the Proportional (P) and Integral (I) components of our controller? Do both these two components contribute to ultrasound image segmentation? Q4: Is the proposed framework applicable to different segmentation tasks and anatomical structures?

### 4.1 EXPERIMENT SETUP

We validate our framework on two distinct and publicly available ultrasound datasets: MEIS , which presents challenges with blurred boundaries in M-mode echocardiography , and TN3K , which involves significant variability in nodule characteristics in B-mode images. For fair comparison, all methods are trained using the SGD optimizer. Our PI controller is configured with a proportional gain $K_{\text{p}} = 2.0$, an integral gain $K_{\text{i}} = 0.1$. We evaluate performance using a suite of standard metrics, including Precision, Recall, Dice Similarity Coefficient (DSC), HD95 and COCO-style mean Average Precision for masks (Mask-mAP) and boxes (Box-mAP). Comprehensive details regarding data preprocessing, augmentation strategies, learning rate schedules, and evaluation protocols are provided in Appendix A.5, A.6, and A.7.

### 4.2 COMPARISON WITH STATE-OF-THE-ART METHODS(Q1,Q4)

We conduct a quantitative comparison of our method against several advanced instance segmentation models, including OSFOMER, CTO, YOLACT, BALANCE, MCADS, and our baseline, RAMEM, on both the MEIS and TN3K datasets.

**Results on MEIS Dataset**. As shown in Table 1, our method achieves the best performance across most key metrics. It obtains the highest Precision (87.63%), DSC (87.55%), and the smallest boundary error measured by HD95 (13.59). While OSFOMER records a marginally higher recall (88.26% vs. our 87.97%), it achieves significantly lower precision (85.77% vs. our 87.63%). In contrast, our method strikes a superior balance, leading to more reliable and accurate segmentation. Instance-level results in Table 3 further show that our method surpasses all baselines in Mask-mAP, Box-mAP, and Avg-mAP. This quantitative superiority is visually corroborated by the qualitative results

Table 1: The results on the MEIS dataset. The best results are highlighted in **bold**, and the second best are underlined.

| Methods | Recall | Precision | DSC | HD95 |
|---|---|---|---|---|
| OSFOMER(ECCV'22) | **88.26±1.81** | 85.77±2.08 | 86.69±1.91 | 17.39±8.68 |
| CTO(IPMI'23) | 87.53±2.05 | 86.05±1.79 | 86.48±1.92 | 17.04±7.51 |
| YOLACT(ICCV'19) | 87.34±1.73 | 87.14±2.16 | 86.92±1.90 | 13.85±5.37 |
| BALANCE(AAAI'25) | 87.48±1.35 | 87.02±1.99 | 86.94±1.67 | 15.48±6.41 |
| MCADS(CVPR'25) | 87.40±2.25 | 86.93±1.83 | 86.79±2.06 | 16.79±9.30 |
| RAMEM(J-BHI'24) | 87.31±2.33 | 87.25±2.01 | 86.91±1.84 | 14.17±5.17 |
| **Ours** | 87.97±2.13 | **87.63±1.97** | **87.55±1.69** | **13.59±5.72** |

Table 2: The results on the TN3K dataset. The best results are highlighted in **bold**, and the second best are underlined.

| Methods | Recall | Precision | DSC | HD95 |
|---|---|---|---|---|
| OSFOMER(ECCV'22) | **91.24±0.52** | 76.86±0.69 | 79.98±0.54 | 63.36±0.73 |
| CTO(IPMI'23) | 85.89±0.37 | 83.09±0.61 | 82.29±0.51 | 42.86±2.02 |
| YOLACT(ICCV'19) | 84.43±0.38 | 80.88±0.57 | 79.88±0.25 | 46.29±2.26 |
| BALANCE(AAAI'25) | 84.12±0.66 | 81.54±0.43 | 80.45±0.49 | 54.02±1.91 |
| MCADS(CVPR'25) | 84.03±0.47 | 82.22±0.37 | 81.17±0.42 | 49.17±3.08 |
| RAMEM(J-BHI'24) | 83.97±0.98 | 84.68±0.82 | 81.91±0.64 | 44.79±2.23 |
| **Ours** | 84.52±0.70 | **85.51±0.57** | **82.55±0.31** | **42.40±1.63** |

in Fig. 2. For the challenging cases, baseline methods like RAMEM and YOLACT often produce fragmented or discontinuous masks. Our model, however, consistently generates predictions that are both complete and precisely aligned with the ground truth boundaries, a direct demonstration of the PI controller's effectiveness in resolving local ambiguities.

**Results on TN3K Dataset**. To validate the adaptability of our framework (Q4), we evaluate it on the TN3K thyroid nodule dataset. The results are detailed in Table 2 and 3. Our approach achieves the top performance in Precision (85.51%), DSC (82.55%), HD95 (42.40), and all mAP-based metrics, culminating in the highest Avg-mAP of 49.20%. This quantitative superiority is visually corroborated by the qualitative results presented in Fig. 3. For challenging cases involving low contrast, irregular shapes, or multiple nodules, baseline methods often yield segmentations with substantial boundary leakage or missed detections. In contrast, our model consistently generates more complete masks that more accurately delineate the ground truth contours, highlighting the PI controller's efficacy in adapting to diverse nodule morphologies. The performance on this distinct anatomical structure and imaging modality highlights the versatility of our control approach, suggesting its potential to resolve persistent errors across different medical ultrasound segmentation tasks. For additional qualitative comparisons, please refer to Appendix A.8.1 and A.8.2.

## 4.3 ABLATION STUDY(Q3)

To deconstruct the contributions of our core components and answer Q3, we conducted an ablation study on both the MEIS and TN3K datasets, as shown in Fig. 4. We compared our full method (Ours) against two variants: one removing the integral term of the PI controller (Ours-PI) and another removing the PI controller and uncertainty loss (Ours-PI-uncertainty).

The results demonstrate that both components are crucial for optimal performance, working in synergy to address different challenges. On the MEIS dataset, with its characteristic blurred boundaries, the integral term showed a more pronounced impact, underscoring its strength in correcting systematic, steady-state errors. On the TN3K dataset, which features high nodule variability, the contributions of both the PI control and the uncertainty were more balanced and closely matched. This highlights their collaborative effectiveness in handling tasks with high data variance.

Table 3: Mask-mAP, Box-mAP and Avg-mAP on MEIS and TN3K datasets. The best results are highlighted in **bold**, and the second best are underlined.

| Methods | MEIS | | | TN3K | | |
|---|---|---|---|---|---|---|
| | Mask-mAP | Box-mAP | Avg-mAP | Mask-mAP | Box-mAP | Avg-mAP |
| OSFOEMER | 54.58±6.39 | 67.73±5.27 | 61.16±5.15 | 48.81±0.71 | 44.24±0.66 | 46.53±0.69 |
| YOLACT | 55.08±5.65 | 66.98±4.99 | 61.03±5.10 | 49.41±0.85 | 44.54±0.79 | 46.97±0.79 |
| RAMEM | 55.67±5.41 | 68.83±4.67 | 62.25±4.83 | 50.80±0.47 | 46.51±0.60 | 48.66±0.52 |
| **Ours** | **57.03±3.87** | **72.67±3.24** | **64.85±2.96** | **51.13±0.34** | **47.27±0.48** | **49.20±0.42** |

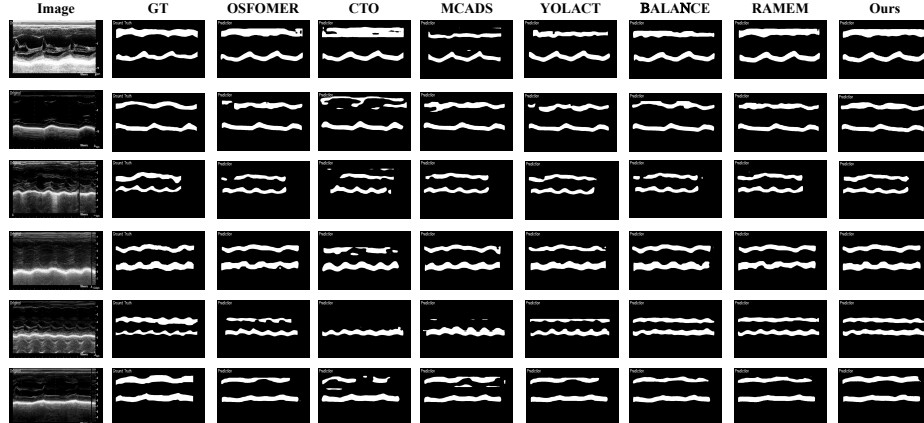

Figure 2: Visualization of segmentation prediction results on the MEIS dataset for different methods.

### 4.4 VISUALIZATION OF STEADY-STATE ERROR ELIMINATION(Q2)

To visually demonstrate that our PI-controlled framework effectively eliminates steady-state error, we design a qualitative comparison experiment, with results shown in Fig. 6. We select several challenging samples where the baseline model performed poorly and compared its error map against that of our method. More visual examples are presented in Appendix A.8.3.

In the figure, each row represents a test case. The first column shows the original input, with red boxes highlighting the most challenging regions. The third column, displaying the baseline's error map, clearly exhibits persistent bright spots within these boxed regions, which correspond to the steady-state error. In comparison, the fourth column shows the error map of

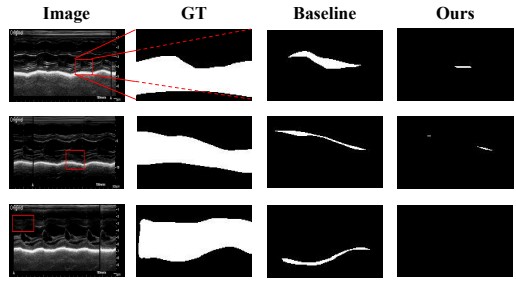

Figure 6: Visualization of segmentation error results on the MEIS dataset for different methods. Red boxes highlight the challenging regions.

our method, where the error in the same regions is significantly suppressed and close to zero. The visualization results show that our PI controller can precisely target and eliminate the steady-state errors that conventional methods fail to resolve, thereby enhancing segmentation reliability. In addition to the qualitative error maps, we also provide a quantitative measurement of steady-state error and visualization showing how model attention changes under PI-Control in Appendix A.10.

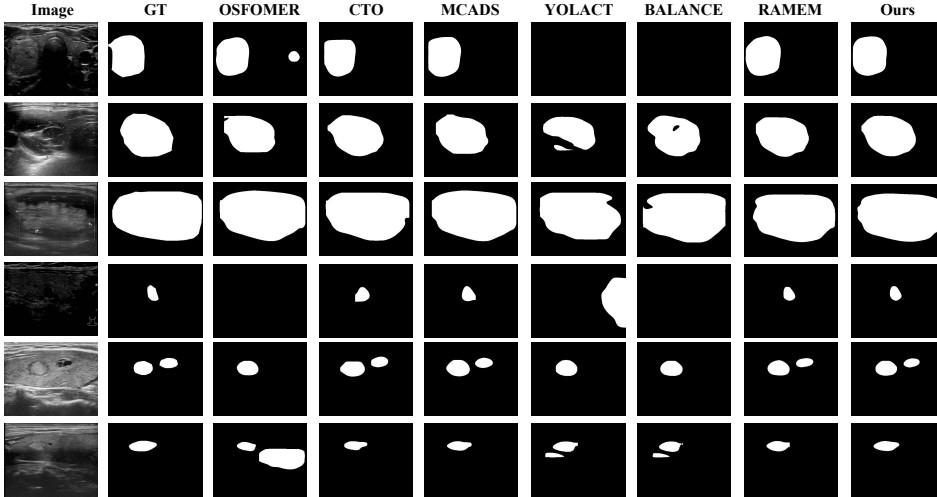

Figure 3: Visualization of segmentation prediction results on the TN3K dataset for different methods.

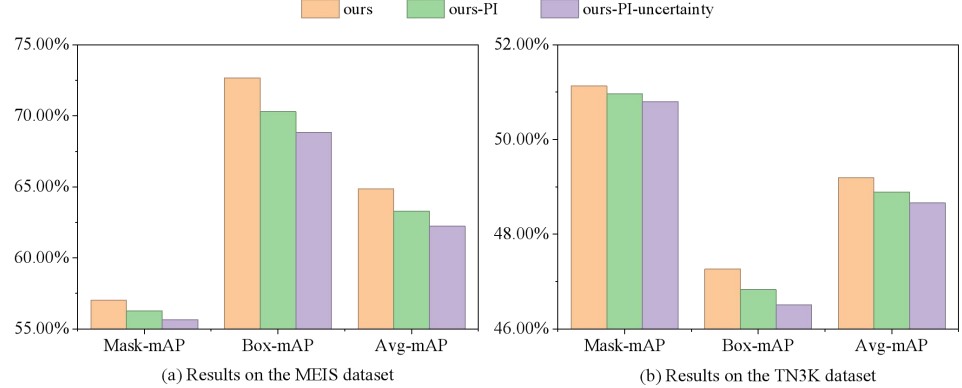

Figure 4: Ablation study. We compare the full model (ours) against a variant without the integral term (ours-PI) and a variant without both the PI control and the uncertainty loss (ours-PI-uncertainty), which represents the RAMEM baseline.

## 4.5 PARAMETER SENSITIVITY ANALYSIS

To investigate the sensitivity of our PI controller to its hyperparameters, the proportional gain ($K_{\mathrm{p}}$) and the integral gain ($K_{\mathrm{i}}$), we conduct parameter sensitivity analysis. As presented in Fig. 5, we vary one parameter while keeping the others fixed to observe the impact on model performance.

The results indicate that the optimal performance is achieved around $K_{\mathrm{p}} = 2.0$ and $K_{\mathrm{i}} = 0.1$. While performance slightly degrades as the parameters deviate from the optimal values, the changes are small. Crucially, the performance across the parameter variation range remains significantly superior to the baseline without the PI controller. This demonstrates that our method exhibits good robustness to hyperparameter selection, facilitating its deployment and tuning in practical applications. Beyond the controller gains, we further analyse the sensitivity with respect to the number of hard pixels $k$, the history length $N$ used in the integral term and likelihood. The results is summarized in Appendix A.11.

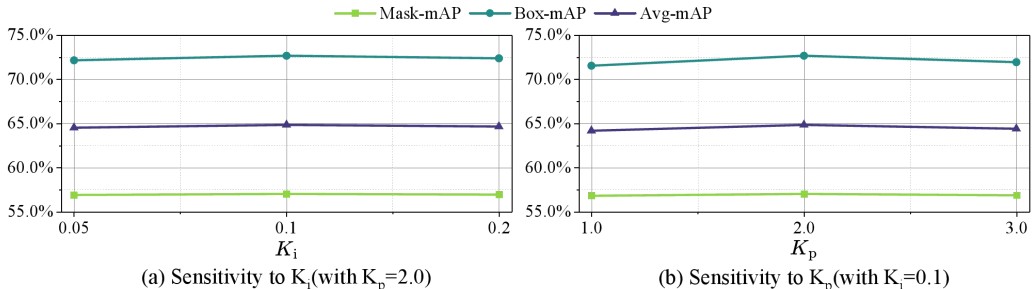

(a) Sensitivity to $K_i$(with $K_p$=2.0)    (b) Sensitivity to $K_p$(with $K_i$=0.1)

Figure 5: Parameter sensitivity analysis for the PI controller gains, $K_p$ and $K_i$, on the MEIS dataset. Plot (a) shows the results of varying the integral gain $K_i$ while holding $K_p = 2.0$. Plot (b) shows the results of varying the proportional gain $K_p$ while holding $K_i = 0.1$

## 5 CONCLUSION

In this paper, we present a novel training framework that rethinks segmentation through the lens of classical control theory. We identify that conventional loss functions are equivalent to proportional controllers, making them incapable of eliminating steady-state errors in challenging segmentation tasks. To address this, we designed a PI controlled uncertainty integrated into the training process. This model-agnostic approach leverages an integral controller to accumulate a memory of historical errors, generating an adaptive corrective signal to resolve stubborn inaccuracies. Experiments on the MEIS and TN3K ultrasound datasets validate our method's effectiveness, showing superior performance over state-of-the-art baselines and visually confirming the elimination of steady-state errors. Moreover, since our training process is explicitly formulated as a closed-loop feedback system, the PI-controlled uncertainty mechanism is conceptually compatible with real-time or continual adaptation scenarios (e.g., test-time adaptation under domain shift). We leave such extensions to online and cross-domain settings as promising directions for future work.

ETHICS STATEMENT

This research utilizes two publicly available medical imaging datasets: MEIS and TN3K. Our study did not involve direct interaction with human subjects, and we did not collect any new patient data. The primary goal of our work is to enhance the accuracy of medical ultrasound image segmentation, which can serve as a beneficial tool to aid clinical diagnosis and decision-making. We acknowledge that any machine learning model trained on specific datasets may carry inherent biases. Future work should investigate the generalizability of our method across diverse patient populations and imaging hardware to ensure fairness and robustness.

REPRODUCIBILITY STATEMENT

To ensure the reproducibility of our work, we provide the following details. **Code:** An anonymized implementation of our method is available at: `https://anonymous.4open.science/r/PI-control-uncertainty-B82C`. **Datasets:** We conduct experiments on two publicly available datasets: MEIS and TN3K. A complete description of the data preprocessing steps for both datasets is provided in Appendix A.5. **Implementation Details:** Our framework is implemented using PyTorch. Full details of the experimental setup, including the learning rate schedule, optimizer, batch size, and data augmentation techniques, are described in Appendix A.7. The key hyperparameters for our proposed PI controller ($K_p$ and $K_i$) are also detailed in Section 4.1, with a sensitivity analysis presented in Section 4.5. **Evaluation:** The evaluation metrics and protocols used to report our results are detailed in Section 4.1 and Appendix A.6.

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

## A  APPENDIX

### A.1  THE USE OF LLMS

The language in this paper is polished by a large language model (LLM) to enhance clarity and readability. The author remains responsible for the final content and academic integrity.

### A.2  BASELINE

Our work builds upon the RAMEM framework, an efficient real-time instance segmentation model adapted from YOLACT for the specific challenges of M-mode echocardiography. Its good performance and high speed make it a suitable backbone for our control-theoretic improvements.

The overall architecture of RAMEM follows the single-stage design of YOLACT, consisting of three core components: a feature extraction backbone, a Feature Pyramid Network (FPN), and parallel prediction heads. Unlike YOLACT, which is based on a traditional ResNet backbone, RAMEM utilizes its proprietary UPANet V2 as the backbone. Its core innovation is the introduction of the Panel Attention mechanism, designed to efficiently build a global receptive field. Since targets in echocardiograms (such as ventricular walls) typically occupy a large portion of the image pixels, the limited local receptive field of traditional CNNs struggles to capture complete structural information. Panel Attention uses a lossless pixel-unshuffle operation to transfer spatial information to the channel dimension, achieving local-to-global attention with low computational overhead and thus resolving this issue. This enables the model to segment large objects more accurately.

RAMEM's prediction heads consist of two parallel sub-networks: the ProtoNet and the Prediction Module. ProtoNet is responsible for generating a series of high-quality, generic prototype masks from a single high-resolution feature map. This process is independent of the number of instances in the image, making it highly computationally efficient. The Prediction Module, on the other hand, operates on multiple feature levels from the FPN, predicting class confidence, bounding boxes, and a set of coefficients for each detected potential instance.

Finally, the segmentation mask for each instance (which is the controlled variable $M_{\text{pred}}$ in our system) is produced by linearly combining the prototype masks from the ProtoNet with the corresponding instance's coefficients predicted by the Prediction Module. This efficient design allows RAMEM to achieve real-time processing while maintaining high accuracy, and our proposed PI controller is designed to regulate and optimize this final output.

### A.3  ALGORITHM

We provide the pseudocode for our proposed PI-Controlled Uncertainty Learning mechanism. Algorithm 1 details the step-by-step process of computing the proportional and integral error terms, generating the modulated error signal, and constructing the final PI-Controlled Uncertainty Loss for each training batch, as described in Section 3.2.2.

---

**Algorithm 1:** PI-Controlled Uncertainty Learning

---

**Input:** $M_{\text{pred}}$: Predicted masks for positive instances at epoch $t$;
   $M_{\text{gt}}$: Ground truth masks corresponding to positive instances;
   $b$: Uncertainty scale parameter;
   $D$: Dictionary storing historical error deques for all instances;
   $K_{\text{p}}, K_{\text{i}}$: Proportional and integral gains;
   $k$: Number of hard pixels
**Output:** $L_{\text{uncertainty}}$: The PI-Controlled Uncertainty Loss

1 Initialize $L_{\text{uncertainty}} \leftarrow 0$;
2 **for** *each instance $i$ in the batch* **do**
3     ▷ *Proportional Term Computation*
4     Compute per-pixel predictive uncertainty: $U^{(i)} \leftarrow 4 \cdot M_{\text{pred}}^{(i)} \cdot (1 - M_{\text{pred}}^{(i)})$;
5     Select the set $\mathcal{H}^{(i)}$ of top-$k$ pixel indices with the highest uncertainty in $U^{(i)}$;
6     Calculate instantaneous instance error: $e_{\text{inst}}^{(i)}(t) \leftarrow \frac{1}{k} \sum_{p \in \mathcal{H}^{(i)}} |M_{\text{gt}}^{(p)} - M_{\text{pred}}^{(p)}|$;
7     $P^{(i)}(t) \leftarrow e_{\text{inst}}^{(i)}(t)$;
8     ▷ *Integral Term computation*
9     Retrieve historical error deque $D^{(i)}$ for instance $i$;
10     Calculate integral term by summing historical errors: $I^{(i)}(t) \leftarrow \sum_{e \in D^{(i)}} e$;
11     ▷ *PI Controller*
12     Compute modulated error: $e_{\text{mod}}^{(i)}(t) \leftarrow K_{\text{p}} \cdot P^{(i)}(t) + K_{\text{i}} \cdot I^{(i)}(t)$;
13     Compute instance loss using Laplace NLL: $L_{\text{inst}}^{(i)} \leftarrow \log(2b^{(i)}) + |e_{\text{mod}}^{(i)}(t)|/b^{(i)}$;
14     $L_{\text{uncertainty}} \leftarrow L_{\text{uncertainty}} + L_{\text{inst}}^{(i)}$;
15     Update history: Append $e_{\text{inst}}^{(i)}(t)$ to deque $D^{(i)}$;
16 **end**
17 **return** $L_{\text{uncertainty}}$;

---

## A.4 DETAILED RELATED WORKS

### A.4.1 RECENT ADVANCES IN MEDICAL IMAGE SEGMENTATION

The state-of-the-art in medical image segmentation has been significantly advanced by models that improve performance through sophisticated architectural and loss-based innovations. A foundational work in real-time instance segmentation is YOLACT (Bolya et al., 2019), which introduced an efficient one-stage paradigm. Building directly upon this, RAMEM (Tseng et al., 2024) adapted the framework for the specific challenges of M-mode echocardiography by enhancing the backbone's receptive field. As RAMEM serves as the direct baseline for our work, we inherit its architectural strengths. Other methods push performance boundaries by enhancing specific network components. For instance, CTO (Lin et al., 2023) introduces an explicit boundary detection operator, OSFORMER (Pei et al., 2022) proposes an efficient one-stage transformer framework to blend local features with long-range context, and MCADS (Wazir & Kim, 2025) rethinks decoder design to better reconstruct fine-grained details. From another perspective, the BALANCE loss (Xu et al., 2025) addresses the issue by re-weighting the loss function to focus on hard-to-classify samples, which predominantly occur at ambiguous boundaries.

While these state-of-the-art methods achieve impressive results, the problem of predictive uncertainty, especially at ambiguous boundaries, remains a core challenge. Some research addresses this using Bayesian methods (Gal & Ghahramani, 2016), Deep Ensembles (Lakshminarayanan et al., 2017), or Evidential Deep Learning (Sensoy et al., 2018) to explicitly quantify uncertainty. However, both the SOTA models and the uncertainty frameworks are limited by their reliance on static losses. They lack a dynamic mechanism to correct for steady-state errors that persist and accumulate during the optimization process itself. This limitation serves as the motivation for our work.

### A.4.2 CONTROL APPROACHES IN DEEP LEARNING

Control theory, particularly principles of feedback and optimization, has been increasingly integrated into deep learning to stabilize training dynamics and enhance performance. For instance, some approaches draw direct parallels between optimizers and controllers. An et al. (2018) formalized this connection by proposing a full Proportional-Integral-Derivative (PID) optimizer. This optimizer incorporates a derivative term (the change in gradient) to anticipate updates and accelerate convergence. In robotics, Jasim Mohamed et al. (2024) developed hybrid control structures that combine neural networks with PID controllers for precise trajectory tracking.

LFB-Net (Girum et al., 2021) formulates segmentation as a recurrent process with an explicit context feedback loop, where the output is encoded and fed back to refine subsequent predictions. Some other works focus on online parameter adaptation. Elkins & Fahimi (2024) models a neural network as a continuous-time dynamical system and applies the Super-Twisting Algorithm (STA) to derive update rules for the network's final layer, thereby guaranteeing error convergence.

However, despite these advances, existing methods often struggle to mitigate severe low contrast, speckle noise, and ambiguous tissue boundaries. Furthermore, the potential of integral control, which is a core component of PI controllers designed to eliminate persistent steady-state errors, remains underexplored in the context of segmentation. Our work designs PI-controlled uncertainty to address the stubborn boundary inaccuracies that many recently proposed segmentation methods struggle to handle.

### A.5 DATASETS

To validate the effectiveness of our proposed method, we conduct experiments on two distinct and publicly available ultrasound datasets: MEIS and TN3K. The two selected datasets represent two distinctly different challenges in medical ultrasound segmentation. MEIS consists of M-mode temporal images with relatively regular structures but blurred boundaries, while TN3K comprises B-mode static images where nodule shape, size, and contrast vary dramatically.

The M-mode Echocardiography of Interventricular Septum (MEIS) dataset is a comprehensive collection of 2,639 M-mode ultrasound images from 923 de-identified subjects, with a resolution of 1024×768 pixels. The dataset covers two standard cardiac views: the Aortic Valve (AV) and the Left Ventricle (LV). For the purposes of our study, we selected a specific subset of 763 images corresponding to the Left Ventricle (LV) view. In this LV view, the key annotated structures are the Interventricular Septum (IVS) and the Left Ventricular Posterior Wall (LVPW). The thicknesses of these structures, along with the Left Ventricular Internal Diameter (LVID), are crucial for deriving key cardiac function indicators such as End-Diastolic Volume (EDV) and End-Systolic Volume (ESV). To isolate the relevant M-mode data for our segmentation task, the images were preprocessed to remove the B-mode components, resulting in a final resolution of 1024×418. The dataset's blurred boundaries and subtle structural details provide a significant challenge for precise segmentation.

The Thyroid Nodule 3000 (TN3K) dataset, provided by the Zhujiang Hospital of Southern Medical University, is highly representative of real-world clinical practice. It comprises 3,493 ultrasound images collected from 2,421 different patients between January 2016 and August 2020. This dataset presents a high degree of complexity, making it an excellent benchmark. The images feature varying dimensions, and approximately 9% of them contain two or more thyroid nodules, increasing the difficulty of the automatic segmentation task. Each image is accompanied by expert-provided, pixel-level annotations. For our experiments, we adhere to the official data split, which designates 2,879 images for the training set and the remaining 614 images for the testing set.

### A.6 EVALUATION METRICS

To provide a comprehensive assessment of our method's performance, we employ a suite of standard metrics for both pixel-level segmentation accuracy and instance-level detection and segmentation quality. Before detailing the metrics, we define the fundamental terms based on a pixel-wise comparison between the predicted segmentation mask ($M_{pred}$) and the ground truth mask ($M_{gt}$): True Positive (TP) is the number of pixels correctly classified as the target object; False Positive (FP) is the number of pixels incorrectly classified as the target object (background predicted as foreground);

False Negative (FN) is the number of pixels incorrectly classified as background (foreground predicted as background).

From these, we derive Precision, which measures the accuracy of positive predictions (TP/(TP+FP)), and Recall, which measures the model's ability to identify all actual positive pixels (TP/(TP+FN)). To balance these two, we use the Dice Similarity Coefficient (DSC), a widely used metric in medical imaging that measures the overlap between the $M_{\text{pred}}$ and the $M_{\text{gt}}$. It is calculated as

$$\text{DSC} = \frac{2 \times |M_{\text{pred}} \cap M_{\text{gt}}|}{|M_{\text{pred}}| + |M_{\text{gt}}|} = \frac{2 \times \text{TP}}{2 \times \text{TP} + \text{FP} + \text{FN}}. \tag{7}$$

In addition to region-overlap metrics, we further evaluate the boundary localization accuracy using the 95th percentile Hausdorff Distance (HD95), a distance-based metric that measures the largest segmentation discrepancy after excluding extreme outliers. Formally, given two boundary point sets $P$ and $G$ extracted from $M_{\text{pred}}$ and $M_{\text{gt}}$, the directed Hausdorff distance is defined as $h(P, G) = \max_{p \in P} \min_{g \in G} \|p - g\|$. The HD95 is computed as the 95th percentile of the symmetric bidirectional distances:

$$\text{HD95} = \text{Percentile}_{95} \Big( \big\{ h(P, G), h(G, P) \big\} \Big), \tag{8}$$

providing a robust indicator of boundary accuracy that is particularly relevant for ultrasound images with ambiguous or fuzzy contours.

For end-to-end instance segmentation performance, we adhere to the standard COCO evaluation protocol, which is based on the mean Average Precision (mAP). This protocol's foundation is the Intersection over Union (IoU) metric, calculated as $\text{IoU}(A, B) = |A \cap B|/|A \cup B|$, which quantifies the overlap between a predicted instance (A) and a ground truth instance (B). The mAP score is obtained by averaging the Average Precision (AP) across a range of IoU thresholds (from 0.5 to 0.95), rewarding models that are accurate at various levels of overlap quality. We report Box-mAP and Mask-mAP, which are calculated using the IoU of bounding boxes and segmentation masks, respectively, to evaluate detection and segmentation quality. Finally, we report the Avg-mAP, which is the mean of Box-mAP and Mask-mAP, providing a single, comprehensive score for overall performance.

### A.7 IMPLEMENTATION DETAILS

All experiments are conducted using PyTorch 1.11.0 and Python 3.8 (Ubuntu 20.04) on a single NVIDIA RTX 4090 GPU with CUDA 11.3. We employ the Stochastic Gradient Descent (SGD) optimizer with an initial learning rate of $1 \times 10^{-3}$, a momentum of 0.9, and a weight decay of $5 \times 10^{-4}$. The learning rate is decayed by a factor of 0.1 at step 13,000. All models are trained for a maximum of 22,000 steps with a batch size of 3. The weights for the overall loss function (Eq. 6) are all set to 1.

We apply several data augmentation techniques during training. These include random adjustments to brightness and contrast, as well as horizontal flipping with a probability of 0.5. All input images are resized to a uniform dimension of 544 × 544, with scale-padding used to maintain the aspect ratio. Pixel values are normalized to the [0, 1] range before being fed into the network. To ensure fair comparison, these enhancement methods are also used in the baseline experiments.

For our proposed PI control mechanism, the key hyperparameters are set as follows: the proportional gain $K_p$ is set to 2.0, and the integral gain $K_i$ is set to 0.1. The controller's memory is configured with a history length of $N_{\text{epochs}} = 5$. For the proportional term's hard pixel mining, we select the top-k pixels, where k was set to 500. Following our two-stage training strategy, the PI-controlled uncertainty loss is activated after the 30th epoch. For fair comparison, all experiments of baseline methods are conducted using the same configuration as the original papers.

### A.8 ADDITIONAL QUALITATIVE RESULTS

This section provides additional qualitative results to supplement the findings presented in the main paper. We showcase more visual comparisons on the MEIS and TN3K datasets to demonstrate the effectiveness of our proposed method on a wider range of challenging cases.

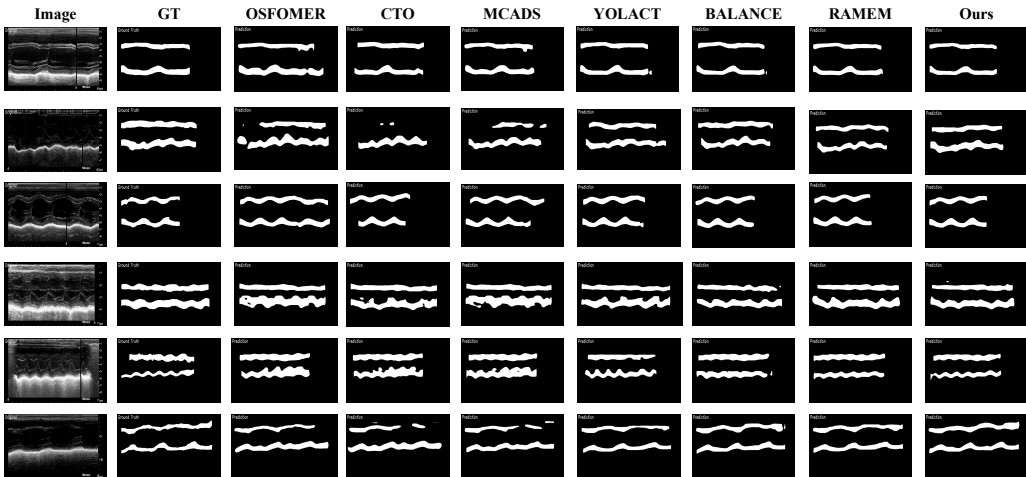

Figure 7: Additional visualization of segmentation prediction results on the MEIS dataset for different methods.

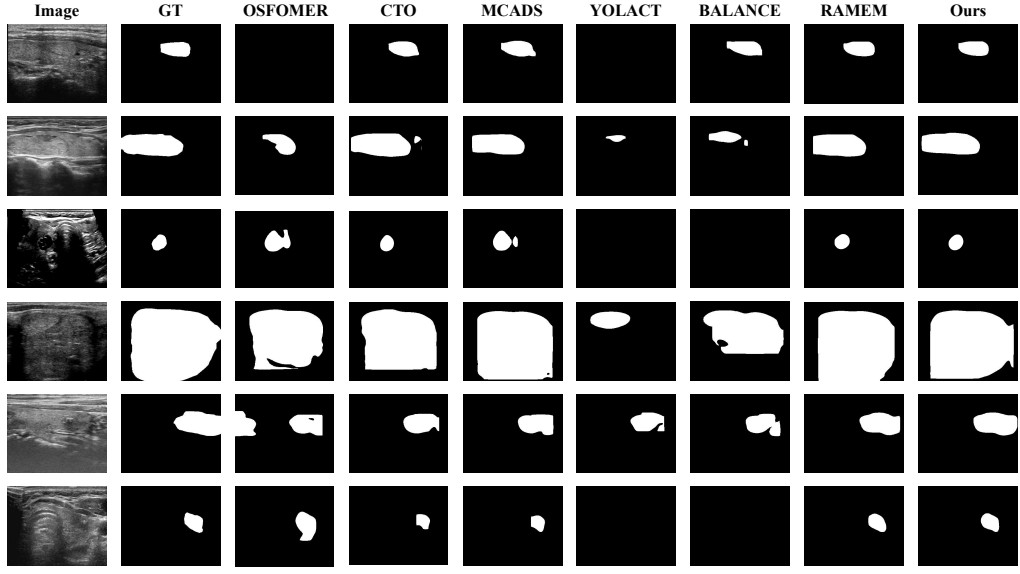

Figure 8: Additional visualization of segmentation prediction results on the TN3K dataset for different methods.

### A.8.1 MORE EXAMPLES ON THE MEIS DATASET

As shown in Fig. 7, we present more segmentation examples on the MEIS dataset, specifically selecting cases where the anatomical contours are ambiguous or discontinuous. Baseline methods, particularly OSFOMER, MCADS, and CTO, tend to produce fragmented and discontinuous masks, failing to capture the integrity of the cardiac structures. While other methods like YOLACT, BALANCE, and RAMEM generate more complete masks, they often struggle with accurately tracing the ground truth contours, resulting in noticeable deviations and less smooth boundaries. Our proposed method consistently maintains structural continuity and provides a much more plausible segmentation. Even in cases where the boundaries are quite ambiguous, our approach yields a result that is substantially more complete and accurate than all baseline methods.

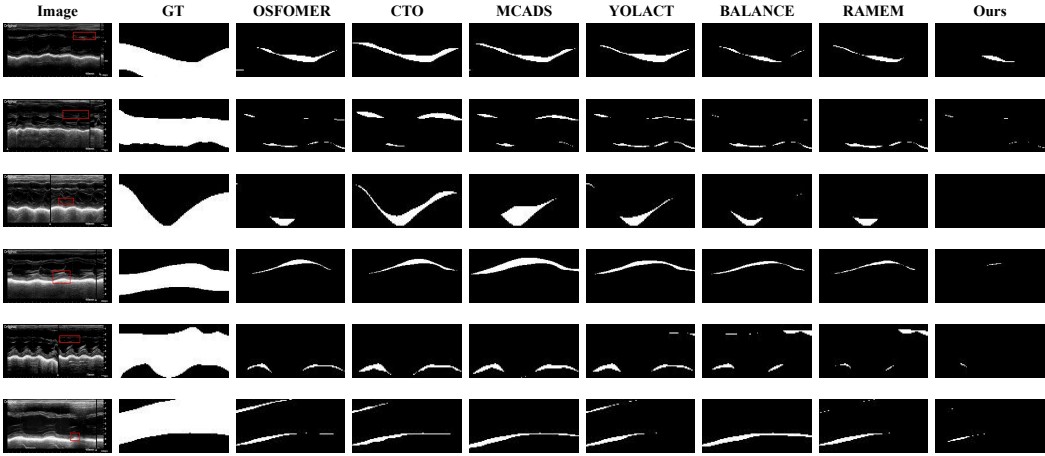

Figure 9: Additional visualization of segmentation error results on the MEIS dataset for different methods. Red boxes highlight the challenging regions.

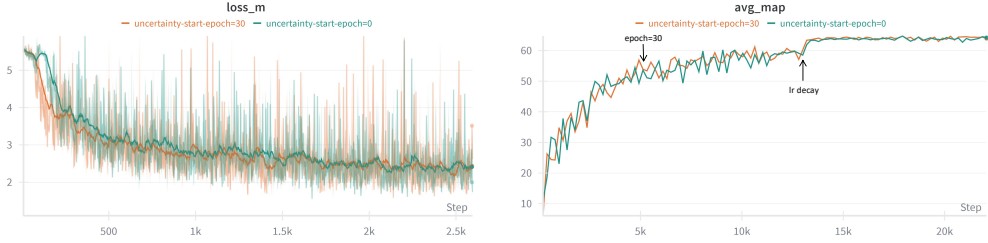

Figure 10: Impact of the uncertainty loss start epoch on training dynamics and performance. (Left) Convergence of mask loss in early training steps. (Right) Evolution of Average mAP throughout the training process.

### A.8.2 MORE EXAMPLES ON THE TN3K DATASET

Fig.8 displays additional results on the TN3K dataset, focusing on nodules with low contrast, irregular shapes, or small sizes. Methods like OSFOMER, BALANCE, and YOLACT often only capture a fraction of the nodule or miss it entirely. Others, such as CTO and RAMEM, may capture the general location but suffer from boundary leakage and fail to segment the precise, often irregular, contours of the nodules. Our method demonstrates a superior ability to adhere to the true nodule boundaries. Even in cases where useful features are extremely sparse, our approach produces a segmentation that is more accurate and faithful to the nodule's true shape than baseline results, making it more reliable for potential clinical assessment.

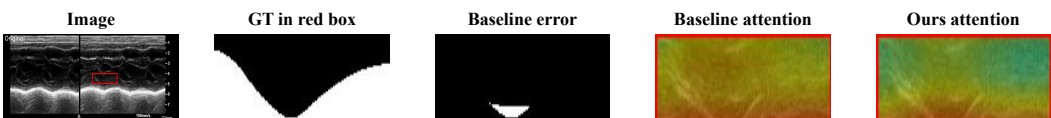

Figure 11: Feature-level attention visualization at epoch 30. The ambiguous ROI is marked by a red box. Left: original image and ground-truth mask. Middle: state-state error in baseline model. Right: JET heatmaps for the baseline and PI-Control.

Table 4: Sensitivity of the PI controller to the number of hard pixels $k$ on MEIS dataset.

| $k$ | 100 | 200 | 400 | 500 | 600 | 800 | 1000 | 2000 |
|---|---|---|---|---|---|---|---|---|
| DSC | 87.38 | 87.41 | 87.49 | 87.55 | 87.60 | 87.58 | 87.54 | 87.47 |
| Mask-mAP | 56.48 | 56.83 | 57.06 | 57.03 | 57.01 | 57.05 | 56.99 | 56.92 |
| Box-mAP | 71.75 | 72.17 | 72.54 | 72.67 | 72.66 | 72.71 | 72.61 | 72.36 |

Table 5: Sensitivity of the PI controller to the history length $N$ on MEIS dataset.

| $N$ | 1 | 3 | 5 | 7 | 9 | 12 |
|---|---|---|---|---|---|---|
| DSC | 87.12 | 87.29 | 87.55 | 87.59 | 87.51 | 87.52 |
| Mask-mAP | 56.36 | 56.64 | 57.03 | 57.08 | 57.04 | 56.98 |
| Box-mAP | 70.88 | 71.30 | 72.67 | 72.72 | 72.58 | 72.60 |

### A.8.3 VISUALIZATION OF STEADY-STATE ERROR ELIMINATION

To supplement the analysis in the main text, Fig. 9 provides a direct visualization of steady-state error elimination on the MEIS dataset. The images in this figure represent the absolute difference between the model's prediction and the ground truth, where white areas indicate segmentation error. The red boxes highlight regions where conventional methods struggle. As shown, all baseline methods exhibit significant residual errors (bright white regions), which are visual manifestations of the steady-state error that persists after training converges. In contrast, the error maps for our method are almost entirely black in these same regions. This provides visual evidence that our PI-controlled framework successfully targets and eliminates the persistent errors that plague standard training paradigms, leading to more reliable and accurate segmentation.

### A.9 STABILITY OF THE PI CONTROLLER AND DELAYED ACTIVATION

Fig. 10 illustrates the impact of the uncertainty loss start epoch on training dynamics. When the PI-controlled uncertainty loss is activated from the beginning of training, the mask loss exhibits slightly stronger oscillations in the early iterations. However, thanks to the bounded integration horizon ($N = 5$) and the small integral gain ($K_i = 0.1$), these fluctuations are quickly damped and the model converges to a similar or even better Avg-mAP compared to the delayed-activation setting.

To avoid unnecessary transient oscillations, we therefore adopt a two-stage training strategy in the main experiments: the model is first optimized using only the static mask loss until a reasonable initial solution is reached, and the PI-controlled uncertainty loss is then activated. Importantly, the closed-loop feedback remains active whenever $L_{\text{uncertainty}}$ is used, so this schedule implements a stabilised PI controller rather than a simple curriculum without feedback.

### A.10 STEADY-STATE ERROR

To quantitatively validate the steady-state error phenomenon, we manually selected a representative ambiguous ROI on the MEIS dataset and measured the percentage of misclassified pixels within this ROI across training epochs. Table 7 reports the results.

The baseline exhibits a clear steady-state error plateau around $10\%$–$12\%$ after approximately 70 epochs, whereas our PI-controlled framework continues to decrease the residual error to below $1\%$, directly demonstrating the effectiveness of the integral term in eliminating persistent segmentation bias.

To examine how the proposed PI-Control influences the model's internal attention patterns during training, we visualize the feature responses at epoch 30 for both the baseline model and the PI-controlled model. We select a representative region of interest (ROI) containing persistent segmentation ambiguity, highlighted by a red bounding box in the original input image. For each model, we extract the corresponding FPN feature map, average it across channels, normalize it to the range $[0, 1]$ using a sigmoid function, and map the result to a JET colormap in which blue denotes low

Table 6: Comparison between Laplace and Gaussian likelihoods on MEIS dataset.

| Likelihood | DSC | Mask-mAP | Box-mAP |
|---|---|---|---|
| Gaussian | 87.18 | 56.68 | 71.05 |
| Laplace (ours) | **87.55** | **57.03** | **72.67** |

Table 7: Steady-state error (%) within an ambiguous ROI over training epochs on MEIS dataset.

| Epoch | 10 | 20 | 30 | 40 | 50 | 60 | 70 | 80 | 90 | 100 |
|---|---|---|---|---|---|---|---|---|---|---|
| RAMEM | 95.29 | 64.28 | 38.37 | 32.75 | 27.15 | 20.44 | 12.27 | 11.29 | 10.77 | 10.83 |
| Ours | 94.96 | 65.93 | 32.75 | 21.86 | 13.38 | 5.82 | 0.69 | 0.43 | 0.38 | 0.41 |

response and red denotes high response. The visualization therefore reflects the relative activation strength and highlights spatial regions that attract stronger feature-level attention.

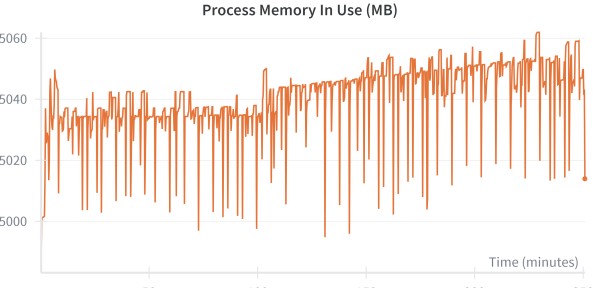

Figure 12: Process memory usage over time.

As shown in Fig. 11, the baseline model produces a diffuse and weak activation distribution: although the ambiguous area shows slightly elevated responses, the attention remains broadly spread and does not strongly emphasize the region responsible for the steady-state error. In contrast, the PI-controlled model exhibits a sharply localized high-response region that aligns precisely with the ambiguous ROI. This concentrated attention indicates that the proportional term identifies uncertain pixels while the integral term accumulates persistent residual errors and amplifies them over time, directing the network to focus on hard-to-correct regions more aggressively. These results provide direct visual evidence that the closed-loop PI mechanism reshapes the internal attention dynamics, enabling the model to attend to regions where errors remain unresolved across epochs.

### A.11 ADDITIONAL ANALYSIS OF PARAMETERS

We conduct a comprehensive parameter analysis to examine how different components of the proposed PI-controlled uncertainty framework affect segmentation performance. Specifically, we study (1) the number of hard pixels $k$ used in the top-k selection, (2) the history length $N$ of the integral term, and (3) the choice of likelihood (Laplace vs. Gaussian) used for modelling the PI-modulated error signal $e_{\mathrm{mod}}(t)$. These parameters directly influence the behaviour of the closed-loop controller and the robustness of the uncertainty modelling.

**Sensitivity to the number of hard pixels** $k$**.** Table 4 reports the performance on MEIS when varying $k$. The results show a smooth trend: performance improves as $k$ increases from 100 to around 400–800, and only slightly degrades when $k$ becomes very large. This is intuitive—the proportional term benefits from a moderate number of challenging pixels but including too many easy pixels weakens its discriminative effect.

**Sensitivity to the history length** $N$**.** Table 5 summarises the effect of varying the integral window size $N$. Performance improves when $N$ increases from 1 to around 5–7, reflecting the benefit of

Table 8: Comparison with Probabilistic U-Net on the MEIS dataset. The best results are highlighted in bold.

| Method | Recall | Precision | DSC | HD95 ↓ |
|---|---|---|---|---|
| Probabilistic U-Net | **88.63± 1.84** | 85.52 ± 2.19 | 86.97 ± 1.82 | 16.25 ± 7.73 |
| RAMEM | 87.31 ± 2.33 | 87.25 ± 2.01 | 86.91 ± 1.84 | 14.17 ± 5.17 |
| Ours | 87.97 ± 2.13 | **87.63± 1.97** | **87.55± 1.69** | **13.59± 5.72** |

accumulating a modest temporal history of residual errors. Larger $N$ does not lead to further gains as excessively long memory weakens the controller's responsiveness, consistent with PI control theory.

**Effect of the likelihood function.** Besides the PI-related parameters, we also compare the Laplace likelihood used in the main paper with a Gaussian alternative that predicts a per-instance variance. As shown in Table 6, the Laplace formulation consistently yields better performance. The heavier-tailed Laplace distribution better models the PI-modulated errors, which may deviate from Gaussian due to the controller's targeted emphasis on hard examples and persistent error regions.

## A.12    MEMORY

Fig. 12 plots the process memory usage over time with the PI controller. The PI-controlled model stores only one scalar error statistic per positive instance for each of the $N$ past epochs. For the MEIS training set with $M = 1174$ instances and $N = 5$, this corresponds to $N \times M = 5870$ scalars.

Empirically, enabling the PI controller increases the peak GPU memory from about 5000 MB to 5060 MB, i.e., less than $1.2\%$ overhead. Since only instance-level scalars are stored, this additional memory cost is independent of the image resolution or dimensionality, indicating that the method can scale to higher-resolution images and even 3D volumes without prohibitive memory growth.

## A.13    ADDITIONAL BASELINE COMPARISON AND CROSS-MODALITY EVALUATION

To further position our method against uncertainty-aware segmentation frameworks, we compare with Probabilistic U-Net(Kohl et al., 2018) on MEIS dataset. As shown in Table 8, our PI-controlled framework achieves higher DSC and Precision and lower HD95, demonstrating that incorporating PI feedback on uncertainty provides more effective error correction than modelling output distributions alone.

We additionally evaluate our framework on the ISIC 2018 dermoscopic lesion segmentation dataset, which differs substantially from ultrasound in imaging physics, contrast patterns and boundary characteristics. Table 9 and 10 reports the results. The consistent gains in DSC, HD95 and Avg-mAP indicate that the proposed PI-controlled optimization is not restricted to ultrasound, but can benefit segmentation under significant domain shifts.

Table 9: Results on the ISIC 2018 dataset. The best results are highlighted in bold.

| Methods | Recall | Precision | DSC | HD95↓ |
|---|---|---|---|---|
| Baseline | **95.47±1.35** | 84.29±1.98 | 87.42±0.41 | 207.56±12.36 |
| Ours | 94.95±0.87 | **86.45±1.66** | **87.93±0.38** | **185.47±9.92** |

Table 10: mAP Results on the ISIC 2018 dataset. The best results are highlighted in bold.

| Methods | Mask-mAP | Box-mAP | Avg-mAP |
|---|---|---|---|
| Baseline | 66.83±3.18 | 60.42±3.74 | 63.62±3.45 |
| Ours | **68.15±2.86** | **63.26±3.51** | **65.70±3.16** |