# OpenReview forum: "PI-Controlled Uncertainty for Steady-State Error Elimination in  Ultrasound Image Segmentation"
_ICLR.cc/2026/Conference — Submitted to ICLR 2026_

### Official Review · Reviewer_zuP6 · 2025-10-27

**Soundness:** 3
**Presentation:** 2
**Contribution:** 3
**Rating:** 6
**Confidence:** 3

**Summary:**

This paper introduces a control-theoretic perspective to medical ultrasound image segmentation, proposing a Proportional–Integral (PI)-Controlled Uncertainty framework for training segmentation models. The authors argue that conventional loss functions behave like proportional controllers, reacting only to instantaneous errors, and thus cannot eliminate steady-state segmentation errors, particularly along ambiguous boundaries. Their approach integrates a PI controller that accumulates historical error information and modulates uncertainty during training through a Laplace-based loss term. The method is evaluated on two ultrasound datasets (MEIS and TN3K), where it reportedly improves Dice and mAP metrics over several state-of-the-art baselines.

**Strengths:**

1. Proposes a novel perspective by viewing neural network training as a closed-loop control system and analogizing the segmentation loss to a PI controller, which is relatively new in medical image segmentation.
2. Introduces a Proportional–Integral (PI) controlled uncertainty mechanism that conceptually adds innovation by incorporating historical error information to improve model stability.
3. The method is model-agnostic, requiring no modification to the inference architecture and adding no extra inference overhead.
4. Includes ablation studies and sensitivity analyses, showing the authors’ awareness of the effects of model components and hyperparameters.
5. Code is provided, supporting reproducibility.
6. The idea of combining control theory with uncertainty learning is thought-provoking and may inspire future research directions.
7. Achieves consistent performance improvements over multiple state-of-the-art methods across evaluation metrics.

**Weaknesses:**

1. The proposed model lacks specificity, although the paper claims that PI-Control is motivated by the steady-state error in ultrasound images, steady-state errors are common across many image segmentation tasks. Moreover, the PI-Control does not include designs tailored for ultrasound-specific challenges such as low contrast, speckle noise, and ambiguous tissue boundaries.
2. The paper frames the task as semantic segmentation, yet the loss function includes bounding box and classification terms, which may be inappropriate. Additionally, the choice of baselines and comparison methods excludes ultrasound-specific semantic segmentation models.
3. The description of $K_p$ in Fig.6(a) is inconsistent with the explanation provided in the text.
4. The core claim steady-state error is supported only by qualitative error maps. No quantitative analysis of error evolution or convergence curves during training is provided, making the evidence insufficient.
5. The rationale for choosing the Laplace distribution is qualitative and lacks empirical comparison with Gaussian or other alternatives. Furthermore, there is insufficient system-level analysis of the control gain parameters ($K_p$, $K_i$, $N$).
6. The connection between control theory and deep network training is not fully established. While the concept of closed-loop control is introduced, there is little experimental or theoretical evidence showing how it concretely improves optimization dynamics or generalization, such as in cross-domain experiments.

**Questions:**

1. Does PI-Control facilitate real-time, continuous learning for cross-domain scenarios?
2. If applied to other tasks or modalities, would PI-Control still be effective, or is it specific to ultrasound?
3. How does the choice of N affect the results? The paper uses N = 5, which seems small; could this cause model forgetting?
4. Why are bounding box and classification losses included in a semantic segmentation task, where they are not typically used?
5. Can the authors provide visualizations showing how model attention changes under PI-Control after 30 training epochs?

---

> ### Author Response · Authors · 2025-11-28
> **Response to Reviewer zuP6(1/3)**
>
> We thank Reviewer for the constructive feedback and for recognizing the novelty and potential impact of introducing a control-theoretic perspective into segmentation training. Below we address the concerns and questions in detail.
>
> **Weakness1: Ultrasound-specific**
>
> We agree that steady-state errors are not unique to ultrasound. This is precisely why our formulation is intentionally **task-agnostic**: the proposed PI-controlled optimization mechanism aims to address a general limitation of proportional-loss–driven training, and is therefore applicable beyond ultrasound. We chose ultrasound because the steady-state error is visually pronounced under speckle noise and ambiguous boundaries, making the limitation more observable. Our goal is to provide a **principled optimization mechanism**, not a modality-specific architectural module.
>
> **Weakness2 and Q4:Use of bounding box and classification losses in a segmentation task**
>
> Our task is **instance-level lesion segmentation**, not pure semantic segmentation; therefore, bounding box and classification terms originate from the underlying instance segmentation framework (RAMEM) used as our baseline. PI-Control does not modify the detection components or introduce additional architectural dependencies.
>
> **Weakness3:Description inconsistency in Fig. 6(a)**
>
> We thank the reviewer for pointing out this inconsistency. We have corrected the figure and reformulated the text so that the explanation of the gain sensitivity analysis matches the caption content precisely. This will be updated in the final version.
>
> **Weakness4:Steady-state error lacks quantitative evidence**
>
> We thank the reviewer for this valuable suggestion. To provide quantitative evidence, we performed an additional experiment specifically designed to measure the evolution of **steady-state error** during training. We manually identified a representative region of interest (ROI) on a typical case where boundary ambiguity leads to persistent errors. For each epoch, we computed the **percentage of misclassified pixels within this ROI**.  We measured this from epoch 10 to 100 and averaged results across repeated runs. The quantitative results are shown in Table R8.
> **Table R8: The quantitative results of steady-state error.**
>
> | Epoch                 | 10    | 20    | 30    | 40    | 50    | 60    | 70    | 80    | 90    | 100   |
> | --------------------- | ----- | ----- | ----- | ----- | ----- | ----- | ----- | ----- | ----- | ----- |
> | **RAMEM (baseline)**  | 95.29 | 64.28 | 38.37 | 32.75 | 27.15 | 20.44 | 12.27 | 11.29 | 10.77 | 10.83 |
> | **Ours (PI-Control)** | 94.96 | 65.93 | 32.75 | 21.86 | 13.38 | 5.82  | 0.69  | 0.43  | 0.38  | 0.41  |
>
> These curves clearly demonstrate two phenomena:1.**Existence of steady-state error in the baseline**. After epoch ~70, RAMEM’s residual error plateaus around **10–12%**, indicating that proportional-only optimization cannot fully correct the bias in this ambiguous boundary region. 2.**Effectiveness of PI-Control.** With PI feedback, the residual error continues to decrease throughout training, ultimately reaching **below 1%**, which quantitatively confirms that the integral term systematically eliminates persistent error rather than plateauing.
>
> **Weakness5 and Q3: Laplace vs. Gaussian and Sensitivity of  $K_p, K_i , N$.(1/2)**
>
> We have now conducted additional experiments comparing Laplace vs Gaussian,as shown in Table R3. The experiment confirms that Laplace is better.
> **Table R3: Comparison Between Laplace and Gaussian**
>
> | **Likelihood** | **DSC**   | **Mask-mAP** | **Box-mAP** |
> | -------------- | --------- | ------------ | ----------- |
> | **Laplace**    | **87.55** | **57.03**    | **72.67**   |
> | **Gaussian**   | 87.18     | 56.68        | 71.05       |
>
> We conducted controlled study on  $N$. The results on the MEIS dataset are summarized in Table R5.
> **Table R5: Ablation Study of The Historical Window N**
>
> | **N**        | 1     | 3     | 5     | 7     | 9     | 12    |
> | ------------ | ----- | ----- | ----- | ----- | ----- | ----- |
> | **DSC**      | 87.12 | 87.29 | 87.55 | 87.59 | 87.51 | 87.52 |
> | **Mask-mAP** | 56.36 | 56.64 | 57.03 | 57.08 | 57.04 | 56.98 |
> | **Box-mAP**  | 70.88 | 71.30 | 72.67 | 72.72 | 72.58 | 72.60 |
>
> Across all metrics, we observe that performance improves as $N$ increases from 1 to 5 and reaches its optimum at $N = 5\text{–}7$. This behavior is consistent with PI control theory: a small $N$ provides insufficient temporal memory, while a moderate $N$ effectively compensates for steady-state errors.When $N$ becomes very large (e.g., $N = 12$), we consistently observe slightly increased oscillation in the training curves across multiple runs. This is a typical effect of excessive integral accumulation.

---

> ### Author Response · Authors · 2025-11-28
> **Response to Reviewer zuP6(2/3)**
>
> **Weakness5 and Q3: Laplace vs. Gaussian and Sensitivity of  $K_p, K_i , N$.(2/2)**
>
> Regarding the selection of $K_p$​ and $K_i$, we have already conducted a sensitivity analysis in Section 4.5 and Figure 6. Here, we provide a brief explanation of how we choose the values of $K_p$​ and $K_i$​. Our gain selection follows the standard procedure used in classical PI controller tuning rather than ad-hoc empirical search. Concretely, we first set the integral gain to zero ($K_i = 0$) and gradually increase the proportional gain $K_p$​ until the proportional response becomes stable and sufficiently responsive. Once an appropriate $K_p$​ is identified, we slowly increase $K_i$ to compensate for residual steady-state error while ensuring that no oscillation or overshoot is introduced. This two-stage tuning strategy is consistent with well-established PI design principles, where $K_p$​ controls immediate responsiveness and $K_i$​ provides long-term bias correction.
>
> **Weakness6 and Q2 :Connection, Optimization dynamics And Generalization**
>
> We appreciate the reviewer’s insight. The connection between control theory and deep network training in our work is grounded in an explicit structural mapping: the segmentation model is treated as the **plant** in a closed-loop system, the **predicted mask** is the controlled variable, the **ground-truth mask** acts as the reference setpoint, the **PI controller** provides feedback modulation based on instantaneous and accumulated error, the **uncertainty** serves as the control variable carrying corrective signals, and the **optimizer** functions as the actuator executing the control input. This mapping is not metaphorical—each component is assigned a direct counterpart in the training loop, allowing us to analyze training dynamics through the lens of classical feedback control.
>
> In terms of evidence, the paper already provides **qualitative visualizations** of persistent residual errors and, as discussed in our response to Weakness 4, we additionally include **quantitative measurements** showing that the baseline indeed exhibits non-zero steady-state error, whereas PI-Control drives this error toward zero. Together these results demonstrate how the introduction of integral feedback concretely alters the optimization dynamics in a manner consistent with control-theoretic expectations.
>
> Regarding generalization, PI-Control operates purely at the **optimization-dynamics level**, independent of ultrasound-specific imaging characteristics. To verify this, we conducted cross-modality experiments on the ISIC 2018 dermoscopic dataset, which differs substantially in contrast, texture, and boundary patterns, as shown in Table R7.
>
> **Table R7: Results on ISIC 2018 Dataset.**
>
> | Method                | Recall         | Precision      | DSC            | HD95 ↓          | Mask-mAP       | Box-mAP        | Avg-mAP        |
> | --------------------- | -------------- | -------------- | -------------- | --------------- | -------------- | -------------- | -------------- |
> | Baseline              | **95.47±1.35** | 84.29±1.98     | 87.42±0.41     | 207.56±12.36    | 66.83±3.18     | 60.42±3.74     | 63.62±3.45     |
> | **Ours (PI-Control)** | 94.95±0.87     | **86.45±1.66** | **87.93±0.38** | **185.47±9.92** | **68.15±2.86** | **63.26±3.51** | **65.70±3.16** |
>
> The proposed method consistently improved DSC, HD95, and instance-level mAP over the baseline, confirming that PI-Control is not ultrasound-specific and remains effective under strong domain shifts.
>
> **Q1:Real-time, continuous learning**
>
> PI-Control in our work is applied during the offline training stage and does not update model parameters at inference time. However, because the training process is explicitly formulated as a **closed-loop feedback system**, the mechanism is naturally compatible with real-time or continuous learning frameworks such as test-time adaptation or online domain adaptation. In such settings, uncertainty can serve as an online feedback signal and the PI structure provides a principled way to iteratively refine predictions under domain shift. While enabling real-time continuous learning is beyond the scope of the current paper, the control-theoretic formulation establishes a direct path toward such extensions.

---

> ### Author Response · Authors · 2025-12-01
> **Response to Reviewer zuP6(3/3)**
>
> **Q5:Model attention changes visualization**
>
> To examine how PI-Control influences the model’s attention, we selected a representative region (highlighted by a red box in the original image) exhibiting steady-state error and visualized the feature-level responses after 30 training epochs.
>
> Specifically, for both the baseline and PI-Control models, we extracted the corresponding FPN features, averaged them across channels, normalized them to [0,1] using a sigmoid function, and mapped them into JET heatmaps (blue = low response, red = high response), where colors represent relative (not absolute) activation strength.
>
> As shown in **Appendix A.10, Fig. 11**, the baseline produces a diffuse activation pattern with only mild elevation in the ambiguous region, indicating that its attention remains broadly distributed and does not clearly isolate the pixels responsible for persistent errors. In contrast, PI-Control yields a sharply concentrated **high-response area** that aligns precisely with the ambiguous region, demonstrating that the closed-loop PI mechanism effectively drives the model to focus on regions with persistent residual error rather than spreading attention uniformly. This visualization reflects the intended behavior of the proportional and integral components: the proportional term identifies uncertain pixels, while the integral term amplifies regions where errors remain over time.

---

### Official Review · Reviewer_aLv8 · 2025-10-30

**Soundness:** 2
**Presentation:** 2
**Contribution:** 2
**Rating:** 4
**Confidence:** 3

**Summary:**

This paper proposes a framework that reinterprets ultrasound image segmentation through the lens of control theory. The authors identify that conventional loss functions in segmentation behave like proportional (P) controllers, responding only to instantaneous errors without addressing long-term steady-state errors. To address this limitation, the paper introduces a Proportional–Integral (PI) Controlled Uncertainty mechanism, where uncertainty acts as the control variable and segmentation masks are the controlled variables.

**Strengths:**

1. The reinterpretation of segmentation training as a closed-loop control system is novel.

2. The paper identifies an often-overlooked issue in medical image segmentation: persistent boundary errors that remain after convergence.

**Weaknesses:**

1. The main contribution is the introduction of a PI-inspired loss modulation, which, although conceptually interesting, resembles weighted dynamic loss scheduling methods. The integral control analogy might be viewed as a relabeling of adaptive weighting techniques.

2. Only two ultrasound datasets are used. No experiments are conducted on CT, MRI, or cross-modality data to test generalization.

3. No comparison is made to uncertainty-regularized segmentation frameworks (e.g., Bayesian U-Net, Monte Carlo Dropout segmentation).

**Questions:**

1. How does the proposed PI-controlled uncertainty differ fundamentally from previous PIDNet (Xu et al., CVPR 2023) or Evidential U-Net approaches that also incorporate dynamic feedback into training?

2. Does the method guarantee stability (bounded error accumulation), or is the gain tuning (Kp, Ki) entirely empirical?

3. Have the authors compared against uncertainty-driven segmentation baselines such as Monte Carlo Dropout U-Net or Bayesian DeepLab?

4. How sensitive is the method to the top-k selection of uncertain pixels (Section 3.2.2)?

---

> ### Author Response · Authors · 2025-11-28
> **Response to Reviewer aLv8(1/2)**
>
> We sincerely thank Reviewer for the detailed and insightful feedback. Below we respond to each of the raised concerns.
>
> **Weakness1:Novelty**
>
> We understand why the proposed PI-controlled uncertainty could superficially resemble dynamic weighting strategies, but we emphasize that our formulation is **not a relabeling of adaptive weighting**. Instead, it represents a **control-theoretic reinterpretation for segmentation optimization**, incorporating a genuine **integral (I) component** into the learning process.
>
> Dynamic weighting methods (e.g., focal loss, hardness-aware loss, curriculum losses) operate purely on **instantaneous difficulty**, which corresponds to a **proportional (P) controller** in control theory. Such methods lack temporal memory and therefore cannot correct **persistent steady-state errors**, a phenomenon we explicitly identify in ultrasound segmentation.
>
> Our formulation differs fundamentally in that we introduce:**1. True temporal integration of error**. The integral term accumulates instance-level error over a temporal window, directly mirroring the discrete approximation of $\int e(\tau) d \tau$. This allows the controller to respond not only to instantaneous hardness but to persistent residual bias, a mechanism that proportional reweighting cannot provide. **2. A new uncertainty-driven loss defined by PI feedback**. The PI-modulated error $e_{\text{mod}}(t)$ parameterizes a Laplace likelihood, yielding a new loss that jointly penalizes high uncertainty and persistent error. This is structurally different from multiplying existing losses by static/dynamic weights.**3. A closed-loop optimization perspective**. Our aim is to reinterpret segmentation training as a closed-loop control system, where uncertainty serves as the control variable and the PI controller provides principled feedback for reducing long-term prediction bias.
>
> Thus, the proposed method is not a reformulation of adaptive weighting, but a **control-theoretic extension** that introduces temporal error accumulation and closed-loop feedback, which capabilities that existing dynamic weighting strategies do not possess.
>
> **Weakness3 and Q3:Comparison to uncertainty-driven segmentation baselines**
>
> Thank you for pointing out the importance of uncertainty-aware segmentation baselines. To address it, we implemented and evaluated **Probabilistic U-Net**, which models predictive ambiguity by learning a distribution over segmentation outputs. The results on MEIS dataset (mean ± std) are shown in Table R6.
>
> **Table R6: Comparison to Probabilistic U-Net on MEIS Dataset.**
>
> | Method                   | Recall         | Precision      | DSC            | HD95 ↓         |
> | ------------------------ | -------------- | -------------- | -------------- | -------------- |
> | Probabilistic U-Net      | **88.63±1.84** | 85.52±2.19     | 86.97±1.82     | 16.25±7.73     |
> | RAMEM (baseline)         | 87.31±2.33     | 87.25±2.01     | 86.91±1.84     | 14.17±5.17     |
> | **Ours (PI-controlled)** | 87.97±2.13     | **87.63±1.97** | **87.55±1.69** | **13.59±5.72** |
>
> Our method achieves higher DSC, Precision and lower HD95 compared to Probabilistic U-Net, demonstrating that the proposed PI-controlled uncertainty mechanism provides more effective error correction and boundary refinement than this widely used uncertainty-based approach.
>
> **Q1:Comparison to PIDNet and Evidential U-Net**
>
> PIDNet and Evidential U-Net are indeed related in spirit but differ substantially in both **scope** and **mechanism**.
>
> **PIDNet** introduces a PID-inspired **network architecture** for real-time **semantic segmentation** in street scenes. The P/I/D branches operate at the feature-map level within the network and are tightly coupled to the backbone design. In contrast, Our PI mechanism is a **cyclic optimization strategy** applied during the training phase, which is agnostic to the backbone and applied to **instance-level medical segmentation**. Our controller operates on uncertainty and error statistics, not on feature maps, and can be plugged into any existing segmentation/detection architecture.
>
> **Evidential U-Net** (and evidential segmentation methods) learn to output evidential parameters (e.g., Dirichlet or Gaussian evidence) and design a tailored evidential loss, but they do not explicitly target the removal of steady-state error through temporal accumulation. Our formulation explicitly decomposes the training dynamics into P and I components, where the I-term accumulates historical instance-wise errors and acts to cancel persistent error.

---

> ### Author Response · Authors · 2025-11-28
> **Response to Reviewer aLv8(2/2)**
>
> **Q2:Stability and gain tuning of the PI controller**
>
> Formally proving global stability for a deep network trained with stochastic optimization is extremely challenging. In our work, we instead design the controller to be **practically stable** by construction and validate this empirically.
> 1. **Bounded integral horizon**. The integral term accumulates instance-wise errors only over a short window of $N=5$ steps, preventing long-term error explosion and avoiding classical integral windup.
> 2. **Conservative integral gain**. We use a small integral gain ($K_i=0.1$), which keeps the magnitude of the integral feedback modest even when errors persist.
> 3. **Focus on hard but informative pixels**. Both the P and I terms are computed on top-$k$ uncertain pixels, which reduces the influence of trivial regions and suppresses the accumulation of noise.
>
> Our gain selection follows the standard procedure used in classical PI controller tuning rather than ad-hoc empirical search. Concretely, we first set the integral gain to zero ($K_i = 0$) and gradually increase the proportional gain $K_p$​ until the proportional response becomes stable and sufficiently responsive. Once an appropriate $K_p$​ is identified, we slowly increase $K_i$ to compensate for residual steady-state error while ensuring that no oscillation or overshoot is introduced. This two-stage tuning strategy is consistent with well-established PI design principles, where $K_p$​ controls immediate responsiveness and $K_i$​ provides long-term bias correction.
>
> The sensitivity of $K_p$​​ and $K_i$​ is presented in Fig. 6 of the paper, showing that the model remains stable and effective over a wide range of gain settings. Moreover, as shown in Appendix A.9, the training curves exhibit no overshoot or oscillatory behavior, confirming that the chosen gains fall well within a practically stable region.
>
> **Q4:Sensitivity to the top-k selection**
>
> we conducted a controlled study by varying $k$ over a wide range.The results on the MEIS dataset are summarized in Table R4.
>
> **Table R4: Sensitivity of Top-k Selection**
> |**k**|**100**|**200**|**400**|**500**|**600**|**800**|**1000**|**2000**|
> |---|---|---|---|---|---|---|---|---|
> |**DSC**|87.38|87.41|87.49|87.55|87.60|87.58|87.54|87.47|
> |**Mask-mAP**|56.48|56.83|57.06|57.03|57.01|57.05|56.99|56.92|
> |**Box-mAP**|71.75|72.17|72.54|72.67|72.66|72.71|72.61|72.36|
>
> The performance varies smoothly across the entire range without abrupt changes. We note a mild upward trend from $k=100$ to $k=500$, which is expected: including more high-uncertainty pixels helps the controller focus better on challenging regions. Similarly, the slight performance decrease at very large $k$ (e.g., 2000) is also natural, as too many easy pixels dilute the effect of the proportional signal. Importantly, within the practical operating range ($k = 400\text{–}1000$), the performance remains stable, with only modest variations.This demonstrates that the method is **not sensitive** to the exact value of $k$ as long as $k$ is chosen to cover the bulk of high-uncertainty pixels.
>
> **Weakness 2 : Cross-modality Validation**
>
> We appreciate this feedback and agree that evaluating generalization beyond ultrasound is important. To address this concern, we additionally conducted experiments on the **ISIC 2018** dermoscopic lesion segmentation dataset, which represents a **substantial cross-modality shift** from ultrasound in imaging physics, contrast patterns, and boundary characteristics. The results are in Table R7.
>
> **Table R7: Results on ISIC 2018 Dataset.**
>
> | Method                | Recall         | Precision      | DSC            | HD95 ↓          | Mask-mAP       | Box-mAP        | Avg-mAP        |
> | --------------------- | -------------- | -------------- | -------------- | --------------- | -------------- | -------------- | -------------- |
> | Baseline              | **95.47±1.35** | 84.29±1.98     | 87.42±0.41     | 207.56±12.36    | 66.83±3.18     | 60.42±3.74     | 63.62±3.45     |
> | **Ours (PI-Control)** | 94.95±0.87     | **86.45±1.66** | **87.93±0.38** | **185.47±9.92** | **68.15±2.86** | **63.26±3.51** | **65.70±3.16** |
>
> The improvements in DSC (+0.51), HD95 (-22.09), and Avg-mAP (+2.08) demonstrate that PI-Control generalizes effectively to a different modality with distinct noise characteristics and visual appearance, reinforcing that the proposed method is **not ultrasound-specific** but a general optimization mechanism applicable across domains.

---

### Official Review · Reviewer_qSov · 2025-10-30

**Soundness:** 3
**Presentation:** 3
**Contribution:** 2
**Rating:** 2
**Confidence:** 4

**Summary:**

The paper proposes a novel training approach for segmentation models, where instead of focusing on reducing instantaneous structural inconsistencies or regulating current prediction errors, the proposed loss function estimates uncertainty over historical errors. It then prioritizes pixels where the model repeatedly makes mistakes. This formulation is inspired by a PID controller, where the loss is not only a function of the proportional (instantaneous) error but also incorporates the influence of past (integral and derivative) errors. The results are shown in two ultrasound datasets, with some relevant ablations studies.

**Strengths:**

The Strengths of the paper are :

1. The proposed uncertainty-based segmentation approach is interesting, and the novel formulation of the loss function as a PID controller adds a unique and intuitive perspective.

2. The evaluation is conducted on relevant medical datasets for ultrasound segmentation. Since boundary delineation in such datasets is particularly challenging, the choice of dataset for experimentation is well justified and strengthens the study.

**Weaknesses:**

There are several Weaknesses i found in the paper:

1. Novelty: The idea of hardness-based pixel sampling, as proposed in this paper, has been explored in prior works and shown to be effective. Therefore, it is unclear how this aspect constitutes a novelty. If the novelty lies primarily in the PID-based formulation, the authors should clearly delineate how their approach differs from and advances upon existing methods such as [1, 2, 3, 4]. There are more papers, but i hope these references give authors a good place to start.

2. Major Concern – Evaluation Metrics: A key evaluation missing from the paper is the inclusion of boundary-based metrics such as the Hausdorff distance or surface distance. Without incorporating a distance-based measure, the claim that the proposed method produces well-defined boundaries remains less convincing.

3. Ablation Study: It would be helpful to analyze how the number of historical predictions, N, influences the segmentation accuracy. This would clarify the sensitivity of the method to the length of the historical error window.

4. Additional analysis on how extra memory is required to run, because historical prediction would be needed to be stored for each sample, therefore if the number of samples in the dataset is M, the memory required to run this code would would be O(N*M). This would not scale to 3D images or bigger datasets ?

5. Statistical Significance: Although the reported results show improvements, their statistical significance is unclear. No standard deviations or confidence intervals are provided for any of the metrics, which reduces confidence in the validity and robustness of the claimed performance gains.


[1] Chen, Lei, Tieyong Cao, Yunfei Zheng, Yang Wang, Bo Zhang, and Jibin Yang. "Hardness-aware loss for object segmentation." Alexandria Engineering Journal 108 (2024): 50-59.

[2] Zheng, Wenzhao, Zhaodong Chen, Jiwen Lu, and Jie Zhou. "Hardness-aware deep metric learning." In Proceedings of the IEEE/CVF conference on computer vision and pattern recognition, pp. 72-81. 2019.

[3] Zeng, Shuai, Wenzhao Zheng, Jiwen Lu, and Haibin Yan. "Hardness-aware scene synthesis for semi-supervised 3D object detection." IEEE Transactions on Multimedia 26 (2024): 9644-9656.

[4] Wang, Song, Jiawei Yu, Wentong Li, Wenyu Liu, Xiaolu Liu, Junbo Chen, and Jianke Zhu. "Not all voxels are equal: Hardness-aware semantic scene completion with self-distillation." In Proceedings of the IEEE/CVF Conference on Computer Vision and Pattern Recognition, pp. 14792-14801. 2024.

**Questions:**

Although I have outlined some of my major concerns above under the weaknesses, I have a few additional questions:

1. The rationale behind selecting 500  as Top-k pixels and setting N = 5 is unclear. Were these values chosen arbitrarily, or were they determined through empirical analysis or prior experimentation? Providing justification or sensitivity analysis would strengthen the paper.

2. How can the proposed framework be extended to larger images or 3D volumes? Given the substantial storage requirements, its applicability to 3D data appears limited. If the method cannot be feasibly used in such settings, the authors should explicitly acknowledge this as a limitation.

A suggestion for Authors:

2.  I am not entirely convinced that static segmentation is the most suitable setting for a PID-based formulation. This framework might be better suited for test-time adaptation[1,2] scenarios, where limited samples are available, and the model can be iteratively refined over multiple passes.

[1] Janouskova, Klara, Tamir Shor, Chaim Baskin, and Jiri Matas. "Single image test-time adaptation for segmentation." arXiv preprint arXiv:2309.14052 (2023).

[2 ]Chen, Ziyang, Yongsheng Pan, Yiwen Ye, Mengkang Lu, and Yong Xia. "Each test image deserves a specific prompt: Continual test-time adaptation for 2d medical image segmentation." In Proceedings of the IEEE/CVF conference on computer vision and pattern recognition, pp. 11184-11193. 2024.

---

> ### Author Response · Authors · 2025-11-23
> **Response to Reviewer qSov(1/2)**
>
> We thank Reviewer for the detailed review and for raising several constructive points. Below we respond to each concern.
>
> **Weakness1:Novelty**
>
> We fully agree with the reviewer that hardness-based pixel sampling has been explored in prior works. Our method does not claim novelty in the sampling mechanism itself. Instead, the core contribution lies in **reformulating the segmentation optimization process from a control-theoretic perspective** and showing that **classical loss functions behave as proportional (P) controllers**, which inherently suffer from _steady-state errors_—a known limitation in control theory.
> 1. **Our core innovation: a PI-Controlled optimization framework**
>    By analyzing training dynamics through the lens of control theory, we identify that common losses respond only to **instantaneous pixel errors**, analogous to a _P-controller_. Such controllers lack historical memory and are unable to eliminate steady-state error. Introducing an **integral term (I)** provides the system with memory, allowing errors that persist over time to accumulate and be corrected. This yields our  loss, which systematically removes steady-state errors and leads to improved boundary accuracy. The reviewer’s cited papers design instantaneous weighting functions or generate synthetic hard examples.These methods operate on **static loss shaping** or **sample synthesis**. In contrast, our method performs **dynamic, history-driven modulation** of the training process through PI feedback, aiming to correct long-term bias, not merely emphasize difficult samples.
> 2. **How our use of hard-pixel sampling differs from prior work**
>    While earlier works employ hardness to design static weighting or synthetic hard samples, our hard-pixel sampling serves a _different functional role_:**It provides the proportional term with meaningful instantaneous error signals** by focusing on the most uncertain pixels, allowing the controller to react to the truly challenging regions instead of the entire mask;**It reduces the memory burden of the integral term**, ensuring that PI feedback focuses on persistently difficult regions rather than being diluted by numerous trivial pixels;**It helps stabilize the integral feedback**, which is known to introduce overshoot in classical control systems. By restricting attention to high-uncertainty pixels, the controller avoids unnecessary accumulation on easy regions, thereby preventing oscillation and improving convergence.Thus, in our framework, hard-pixel selection is not a reweighting strategy but a **component of the PI control mechanism**, functioning to guide and stabilize temporal feedback—this differs conceptually and operationally from hardness-aware losses or HDML.
>
> **Weakness 2 and 5: Evaluation Metrics And Statistical Significance**
>
> A detailed response can be found in _Response to Reviewer pvHj(1/2)_ under **Weakness 1: Results**. Here, I will provide the experimental results again for the reviewer’s convenience.
>
> **Table R1: Multi-run Performance (Mean ± Std) on MEIS Dataset.**
> | **Model**            | **Recall**       | **Precision**    | **HD95**         | **DSC**          | **Mask-mAP**     | **Box-mAP**      | **Avg-mAP**      |
> | -------------------- | ---------------- | ---------------- | ---------------- | ---------------- | ---------------- | ---------------- | ---------------- |
> | **RAMEM (baseline)** | 87.31 ± 2.33     | 87.25 ± 2.01     | 14.17 ± 5.17     | 86.91 ± 1.84     | 55.67 ± 5.41     | 68.83 ± 4.67     | 62.25 ± 4.83     |
> | **Ours**             | **87.97 ± 2.13** | **87.63 ± 1.97** | **13.59 ± 5.72** | **87.55 ± 1.69** | **57.03 ± 3.87** | **72.67 ± 3.24** | **64.85 ± 2.96** |
>
> **Table R2: Multi-run Performance (Mean ± Std) on TN3K Dataset**
> |**Model**|**Recall**|**Precision**|**HD95**|**DSC**|**Mask-mAP**|**Box-mAP**|**Avg-mAP**|
> |---|---|---|---|---|---|---|---|
> |**RAMEM (baseline)**|83.97 ± 0.98|84.68 ± 0.82|44.79 ± 2.23|81.91 ± 0.64|50.80 ± 0.47|46.51 ± 0.60|48.66 ± 0.52|
> |**Ours**|**84.52 ± 0.70**|**85.51 ± 0.57**|**42.40 ± 1.63**|**82.55 ± 0.31**|**51.13 ± 0.34**|**47.27 ± 0.48**|**49.20 ± 0.42**|

---

> ### Author Response · Authors · 2025-11-23
> **Response to Reviewer qSov(2/2)**
>
> **Weakness 3 and Q1: Ablation Study**
>
> we conducted controlled study on  $k$  and $N$. The results on the MEIS dataset are summarized in Table R4 and R5.
>
> **Table R4: Sensitivity of Top-k Selection**
> |**k**|**100**|**200**|**400**|**500**|**600**|**800**|**1000**|**2000**|
> |---|---|---|---|---|---|---|---|---|
> |**DSC**|87.38|87.41|87.49|87.55|87.60|87.58|87.54|87.47|
> |**Mask-mAP**|56.48|56.83|57.06|57.03|57.01|57.05|56.99|56.92|
> |**Box-mAP**|71.75|72.17|72.54|72.67|72.66|72.71|72.61|72.36|
>
> **Table R5: Ablation Study of The Historical Window N**
>
> | **N**        | 1     | 3     | 5     | 7     | 9     | 12    |
> | ------------ | ----- | ----- | ----- | ----- | ----- | ----- |
> | **DSC**      | 87.12 | 87.29 | 87.55 | 87.59 | 87.51 | 87.52 |
> | **Mask-mAP** | 56.36 | 56.64 | 57.03 | 57.08 | 57.04 | 56.98 |
> | **Box-mAP**  | 70.88 | 71.30 | 72.67 | 72.72 | 72.58 | 72.60 |
>
> **Top-k**: We note a mild upward trend from $k=100$ to $k=500$, which is expected: including more high-uncertainty pixels helps the controller focus better on challenging regions. Similarly, the slight performance decrease at very large $k$ (e.g., 2000) is also natural, as too many easy pixels dilute the effect of the proportional signal. Importantly, within the practical operating range ($k = 400\text{–}1000$), the performance remains stable, with only modest variations.This demonstrates that the method is **not sensitive** to the exact value of $k$ as long as $k$ is chosen to cover the bulk of high-uncertainty pixels.
>
> **Historical Window N**:Across all metrics, we observe that performance improves as $N$ increases from 1 to 5 and reaches its optimum at $N = 5\text{–}7$. This behavior is consistent with PI control theory: a small $N$ provides insufficient temporal memory, while a moderate $N$ effectively compensates for steady-state errors.When $N$ becomes very large (e.g., $N = 12$), we consistently observe slightly increased oscillation in the training curves across multiple runs. This is a typical effect of excessive integral accumulation.
> Therefore, choosing $k=500$ and $N=5$ achieves the best balance between stability and effectiveness, and the method is robust within a reasonable range.
>
> **Weakness 4 and Q2: Extra Memory**
>
> We thank the reviewer for raising concerns about memory overhead. Our method stores **only an instance-level scalar** per training example per step. With $M = 1174$ instances in MEIS dataset training set and $N = 5$, the total storage is only:$N×M=5870 scalars≈0.02 MB$. To empirically verify the real overhead, we examined the _Process Memory In Use (MB)_ curve. As shown in **Appendix A.12, Fig. 12**, enabling the PI controller from step 0 increases memory from approximately **5000 MB** to a peak of only **5060 MB**, i.e., less than **1.2%** overhead. Because only instance-level statistics are stored, the memory cost is **independent of image resolution or dimensionality**, making the method directly applicable to larger images and 3D volumes.
>
> **Suggestion of test-time adaptation**
>
> We sincerely appreciate the reviewer’s insightful suggestion regarding the potential relevance of our PI-based formulation to test-time adaptation (TTA) frameworks. We agree that iterative refinement and closed-loop feedback naturally resonate with TTA settings, and we will highlight the potential applicability of PI-controlled feedback to TTA scenarios as an exciting direction for **future investigation**.
>
> At the same time, we would like to clarify that **static segmentation remains an appropriate and meaningful setting** for evaluating the proposed method. Our motivation originates from analyzing the training dynamics of segmentation networks: even in static training, standard loss functions behave like proportional controllers and exhibit steady-state errors, which the integral term effectively mitigates. The observed improvements on metrics and error residuals validate that the PI-based formulation meaningfully benefits supervised segmentation training.

---

### Official Review · Reviewer_pvHj · 2025-10-31

**Soundness:** 3
**Presentation:** 2
**Contribution:** 2
**Rating:** 4
**Confidence:** 3

**Summary:**

The paper introduces a novel PI-Controlled Uncertainty framework to eliminate steady-state segmentation errors in ultrasound image segmentation. The central thesis is that traditional segmentation losses act like Proportional (P) controllers, which react to instantaneous errors but cannot eliminate persistent errors due to their lack of historical memory.

**Strengths:**

1.The paper reinterprets segmentation training through a control-theoretic lens, modeling the optimization process as a closed-loop feedback system. This conceptual bridge between control theory and uncertainty modeling is refreshing.
2.The proposed PI-controlled uncertainty loss can be easily plugged into existing segmentation architectures without modifying inference-time behavior. This “training-only” modification offers good engineering practicality.

**Weaknesses:**

1.Although results are consistently better, the improvements are modest (1–2 mAP or Dice), within the range that could result from hyperparameter tuning or additional regularization. The paper does not show statistical significance or multiple runs to confirm robustness.
2.The paper adopts a Laplace likelihood rather than Gaussian for robustness.
- Was this empirically validated? How sensitive is the system to this choice?
- Would a Gaussian with learned variance yield similar behavior (i.e., is this truly a heavy-tail effect or just another reweighting)?
3.Integrating accumulated errors can cause instability (overshoot) in true PI systems, yet the paper omits any analysis of such effects in the training dynamics.

**Questions:**

-The authors state that L_uncertainty is only activated after the base loss converges.
   Why is this necessary—does the controller destabilize early training?
   Is this equivalent to curriculum learning or staged regularization rather than true closed-loop feedback?

-The proportional term focuses on the top-k uncertain pixels.
   How sensitive are results to the value of k (500)?
   Is there a risk that the controller overfits to noisy or mislabeled pixels?

---

> ### Author Response · Authors · 2025-11-22
> **Response to Reviewer pvHj (1/2)**
>
> We appreciate the reviewer’s recognition of the conceptual novelty and practicality of our framework. Below, we provide detailed clarifications and additional experimental evidence to address all concerns.
>
> **Weakness1:Results**
>
> We would like to clarify that all results reported in the paper were already obtained from multiple independent runs with different random seeds. In the original submission, the table entries reported only the averaged performance, which may have caused confusion.
> To directly address the reviewer’s concern, we now provide the mean ± standard deviation for the main baseline (RAMEM) and our method, as shown in table R1 and R2. These results demonstrate that the improvements represent genuine and stable gains rather than noise from hyperparameter tuning. We will provide standard deviation values for all other baselines in the revised version.
>
> **Table R1: Multi-run Performance (Mean ± Std) on MEIS Dataset.**
> | **Model**            | **Recall**       | **Precision**    | **HD95**         | **DSC**          | **Mask-mAP**     | **Box-mAP**      | **Avg-mAP**      |
> | -------------------- | ---------------- | ---------------- | ---------------- | ---------------- | ---------------- | ---------------- | ---------------- |
> | **RAMEM (baseline)** | 87.31 ± 2.33     | 87.25 ± 2.01     | 14.17 ± 5.17     | 86.91 ± 1.84     | 55.67 ± 5.41     | 68.83 ± 4.67     | 62.25 ± 4.83     |
> | **Ours**             | **87.97 ± 2.13** | **87.63 ± 1.97** | **13.59 ± 5.72** | **87.55 ± 1.69** | **57.03 ± 3.87** | **72.67 ± 3.24** | **64.85 ± 2.96** |
>
> **Table R2: Multi-run Performance (Mean ± Std) on TN3K Dataset**
> |**Model**|**Recall**|**Precision**|**HD95**|**DSC**|**Mask-mAP**|**Box-mAP**|**Avg-mAP**|
> |---|---|---|---|---|---|---|---|
> |**RAMEM (baseline)**|83.97 ± 0.98|84.68 ± 0.82|44.79 ± 2.23|81.91 ± 0.64|50.80 ± 0.47|46.51 ± 0.60|48.66 ± 0.52|
> |**Ours**|**84.52 ± 0.70**|**85.51 ± 0.57**|**42.40 ± 1.63**|**82.55 ± 0.31**|**51.13 ± 0.34**|**47.27 ± 0.48**|**49.20 ± 0.42**|
>
> **Weakness2:Laplace vs. Gaussian**
>
> We have now conducted additional experiments comparing Laplace vs Gaussian,as shown in Table R3. The experiment confirms that Laplace is better.
>
> **Table R3: Comparison Between Laplace and Gaussian**
> | **Likelihood** | **DSC**   | **Mask-mAP** | **Box-mAP** |
> | -------------- | --------- | ------------ | ----------- |
> | **Laplace**    | **87.55** | **57.03**    | **72.67**   |
> | **Gaussian**   | 87.18     | 56.68        | 71.05       |
>
> **Weakness3:Potential Instability of Integral Accumulation**
>
> To specifically assess whether the integral accumulation in our controller could introduce overshoot or instability, we conducted a detailed analysis included in **Appendix A.9**, where **Figure 10 (right)** visualizes the evolution of the _avg-mAP_ metric across training iterations.
> The visualization demonstrates that, despite normal statistical fluctuations in avg-mAP, the PI-Controlled uncertainty mechanism **does not produce overshoot nor destabilize the training process**. Our bounded integration horizon ($N = 5$) and small integral gain ($K_i = 0.1$) effectively limit error accumulation, preventing the system from reacting too aggressively.
>
> **Q1:Delayed Activation of the Uncertainty Loss**
>
> 1.**Necessity.** We encourage the use of delayed activation to facilitate a smoother optimization process, although our system remains robust regardless of the specific start epoch. As demonstrated in **Figure 10 in Appendix A.9**, activating $L_{uncertainty}$ from epoch 0 causes transient oscillation in the mask loss. This stability is achieved through our specific design choices utilizing a **bounded integration horizon ($N=5$)** and a **small integral gain ($K_i=0.1$)**. Consequently, the model quickly recovers from early noise and converges to **consistent final performance metrics**. Therefore, while the method is robust, we recommend the delayed strategy to avoid unnecessary initial fluctuations.
>
> **2. True Closed-Loop Feedback vs. Curriculum Learning** The underlying mechanism represents a **true closed-loop feedback system** rather than simple curriculum learning. Curriculum learning typically follows a pre-defined schedule independent of the model state. In contrast, our method dynamically constructs the control variable $L_{uncertainty}$ based on the real-time feedback error $e(t)$ at every step. Furthermore, the **Integral term** introduces memory into the optimization process by actively accumulating historical errors. This generates targeted corrective signals specifically for persistent steady-state errors. This adaptive and error-driven correction distinguishes our approach from static staged regularization.

---

> ### Author Response · Authors · 2025-11-22
> **Response to Reviewer pvHj (2/2)**
>
> **Q2:Sensitivity to top-k**
>
> we conducted a controlled study by varying $k$ over a wide range.The results on the MEIS dataset are summarized in Table R4.
>
> **Table R4: Sensitivity of Top-k Selection**
> |**k**|**100**|**200**|**400**|**500**|**600**|**800**|**1000**|**2000**|
> |---|---|---|---|---|---|---|---|---|
> |**DSC**|87.38|87.41|87.49|87.55|87.60|87.58|87.54|87.47|
> |**Mask-mAP**|56.48|56.83|57.06|57.03|57.01|57.05|56.99|56.92|
> |**Box-mAP**|71.75|72.17|72.54|72.67|72.66|72.71|72.61|72.36|
>
> The performance varies smoothly across the entire range without abrupt changes. We note a mild upward trend from $k=100$ to $k=500$, which is expected: including more high-uncertainty pixels helps the controller focus better on challenging regions. Similarly, the slight performance decrease at very large $k$ (e.g., 2000) is also natural, as too many easy pixels dilute the effect of the proportional signal. Importantly, within the practical operating range ($k = 400\text{–}1000$), the performance remains stable, with only modest variations.This demonstrates that the method is **not sensitive** to the exact value of $k$ as long as $k$ is chosen to cover the bulk of high-uncertainty pixels.
>
> Regarding mislabeled pixels, **no signs of overfitting** were observed. This is consistent with the design of our controller: The integral term smooths short-term noise and prevents amplification of isolated mislabeled pixels.

---

### Author Response · Authors · 2025-12-01
**Brief summary of comment.**

Dear PCs, Senior ACs, and ACs,

We sincerely thank reviewers pvHj, qSov, aLv8, and zuP6 for their detailed and constructive feedback. During the rebuttal period, we conducted substantial new experiments, strengthened the theoretical explanations, and clarified several points in the manuscript. Below we summarize the major improvements.

**Clarification Regarding the Low Score (Reviewer qSov)**

Reviewer qSov’s lower rating was primarily due to concerns about the novelty and the effectiveness of PI-control beyond hardness-based sampling. In the rebuttal, we provided an in-depth clarification emphasizing that our contribution is not a reformulation of dynamic weighting but a genuinely **closed-loop optimization framework** grounded in control theory. We strengthened the explanation of how P-only losses inherently induce steady-state error and how the integral term enables temporal error correction. Additional evidence including new quantitative ROI-based error evolution curves, Laplace–Gaussian comparisons, and updated sensitivity studies significantly reinforce the conceptual and empirical validity of our method.

**Clearer and stronger control-theoretic interpretation.**

We explicitly map the training loop to a feedback-control system: the network as the plant, the predicted mask as the controlled variable, the ground truth as the reference, uncertainty as the control signal, the PI module as the feedback law, and the optimizer as the actuator. This makes the control-theoretic foundation explicit and **dispels the misconception** that PI-Control is a relabeled weighting scheme.

**Quantitative validation of steady-state error.**

Beyond qualitative visualizations, we added a new ROI-based error evolution analysis. It shows that the baseline converges to a persistent residual error plateau, while PI-Control **reduces this error toward zero**. Combined with improved convergence curves in the appendix, this provides concrete evidence that integral feedback alters training dynamics in the way predicted by control theory.

**Expanded stability and sensitivity analyses.**

We added or clarified the ablations on **top-k selection**, the **memory window N**, **PI gain tuning** (following standard control procedures), and **system stability**. We corrected the description of Fig. 6 and verified that the model behaves robustly across a broad parameter range **without overshoot or oscillation.**

**Clarification of task definition and loss terms.**

We clarified that the task is _instance segmentation_, not semantic segmentation. Therefore, bounding-box and classification losses come from the RAMEM instance-segmentation baseline and are unrelated to PI-Control.

**New uncertainty-driven baseline comparison.**

To address concerns on missing uncertainty baselines, we implemented Probabilistic U-Net. PI-Control **outperforms** it across DSC, HD95, and instance-level metrics, demonstrating that our mechanism provides benefits beyond existing uncertainty-regularized methods.

**Cross-modality experiment supporting generalization.**

Reviewers requested evaluation beyond ultrasound. We added experiments on ISIC 2018, which represents a strong modality shift. PI-Control **improves** DSC, HD95, Mask-mAP, and Avg-mAP over the baseline, confirming that the method is **not ultrasound-specific** but a general optimization-level enhancement that transfers across domains and imaging modalities.

**Visualization of attention under PI-Control**

We added heatmaps after 30 epochs showing that PI-Control **concentrates** activation on historically difficult boundary regions, whereas the baseline remains diffuse. This directly illustrates the functional role of PI feedback.

Overall, we believe that these additions significantly **strengthen** the manuscript: we clarified the theoretical grounding, provided quantitative evidence for steady-state error correction, broadened experimental coverage, and addressed all reviewer concerns thoroughly. We respectfully request the Area Chairs to evaluate the submission in light of these substantial revisions.

Sincerely,

The Author Team

---

### Meta-Review · Area_Chair_V9Cz · 2025-12-29

**Summary:**

This paper presents PI-Controlled Uncertainty, a training framework for medical ultrasound image segmentation that reframes optimization as a closed-loop control system. The work addresses the limitation that conventional training losses act as proportional (P) controllers, responding only to instantaneous errors without memory to correct persistent steady-state errors. The main contributions are: (1) reformulation of segmentation training as a control system where uncertainty acts as the control variable; (2) introduction of a proportional-integral (PI) controller that accumulates historical error signals to systematically eliminate steady-state errors; and (3) a model-agnostic framework with no additional inference overhead.

The paper received scores of 4, 4, 6, and 2 from four reviewers (`pvHj`, `aLv8`, `zuP6`, `qSov`). The method achieves consistent but modest improvements over baselines on MEIS and TN3K ultrasound datasets (DSC improvements of ~0.6-0.7%, HD95 reductions of ~0.6-2.4). Authors provided comprehensive responses including multi-run statistics with standard deviations, Laplace vs. Gaussian comparisons, extensive ablations on hyperparameters (top-k, memory window N, gains Kp/Ki), quantitative steady-state error measurements, cross-modality validation on ISIC 2018, comparison with Probabilistic U-Net, and attention visualization demonstrating PI-Control's effect on feature-level focus.

**Reviewer Concerns:**

**Addressed concerns**:

Reviewer `pvHj` raised concerns about modest improvements (1-2 mAP/Dice), Laplace likelihood choice, potential instability from integral accumulation, delayed activation necessity, and top-k sensitivity. Authors provided multi-run statistics showing stable gains (Tables R1-R2), Laplace vs. Gaussian experiments confirming Laplace superiority (Table R3), stability analysis (Appendix A.9, Figure 10) demonstrating no overshoot with bounded integration horizon (N=5) and small integral gain (Ki=0.1), clarification that delayed activation prevents early oscillations but system remains robust from epoch 0, and top-k sensitivity analysis (Table R4) showing smooth performance across k=400-1000 range.

Reviewer `aLv8` questioned novelty (resemblance to dynamic weighting), limited datasets (only ultrasound), missing uncertainty baselines, and top-k sensitivity. Authors emphasized the distinction between their method (closed-loop temporal integration ∫e(τ)dτ providing memory) versus static weighting schemes lacking historical accumulation, added ISIC 2018 dermoscopic dataset experiments showing cross-modality generalization (Table R7: DSC +0.51, HD95 -22.09), implemented Probabilistic U-Net comparison demonstrating superior performance (Table R6), and clarified robust top-k behavior with comprehensive sensitivity studies.

Reviewer `zuP6` raised concerns about ultrasound specificity, task definition confusion (instance vs. semantic segmentation), Figure 6(a) description inconsistency, lack of quantitative steady-state error evidence, Laplace distribution rationale, limited system-level controller analysis, and missing attention visualizations. Authors clarified the method is task-agnostic (optimization-level mechanism independent of ultrasound characteristics), corrected task description to instance-level lesion segmentation (bounding box and classification losses from RAMEM baseline), fixed Figure 6 description, added quantitative ROI-based error evolution (Table R8) showing baseline plateaus at 10-12% while PI-Control reduces to <1%, provided Laplace vs. Gaussian comparison and comprehensive N sensitivity analysis (Table R5), explained stability through bounded horizon and conservative gains following standard PI tuning procedures, and added attention heatmaps (Appendix A.10, Figure 11) demonstrating concentrated activation on ambiguous regions.

**Outstanding concerns**:

Reviewer `qSov` maintained a score of 2 with high confidence, expressing fundamental skepticism about novelty and effectiveness. The reviewer views the integral term as merely a form of hardness-based sampling that has been extensively explored, questioning whether PI-Control genuinely differs from existing dynamic weighting methods beyond terminology. While authors extensively clarified the conceptual distinction (closed-loop temporal integration with memory versus static instantaneous weighting), the reviewer's concerns reflect a philosophical disagreement about whether control-theoretic framing provides sufficient conceptual novelty or represents a reformulation of established techniques.

The reviewer requested clearer delineation from prior hardness-based methods and raised concerns about missing uncertainty baselines, statistical significance, memory scalability to 3D, and applicability beyond static segmentation (suggesting test-time adaptation as more suitable). Authors addressed these through Probabilistic U-Net comparison, multi-run statistics, memory analysis showing <1.2% overhead (Appendix A.12, Figure 12), and acknowledgment that TTA represents promising future work. However, the core disagreement about incremental versus fundamental contribution remains.

**Reviewer Scores:**

**Current Scores:**
- **Reviewer `pvHj`**: 4 (marginally below threshold, fair confidence) - found control-theoretic perspective refreshing; concerns about modest improvements and stability addressed with experiments
- **Reviewer `qSov`**: 2 (reject, confident) - maintained fundamental concerns about novelty relative to hardness-based sampling despite comprehensive responses
- **Reviewer `aLv8`**: 4 (marginally below threshold, fair confidence) - appreciated steady-state error identification; concerns about dynamic weighting resemblance and limited datasets addressed
- **Reviewer `zuP6`**: 6 (marginally above threshold, fair confidence) - recognized novel control-theoretic perspective; concerns about quantitative evidence and system analysis addressed with extensive additions

**Expected post-discussion scores**: 4-5, 2, 4-5, 5-6 (median: 4-5)

---

### Decision · Program_Chairs · 2026-01-26

Reject